



# Comparison of past and future simulations of ENSO in CMIP5/PMIP3 and CMIP6/PMIP4 models

Josephine R. Brown[1], Chris M. Brierley[2], Soon-Il An[3], Maria-Vittoria Guarino[4], Samantha Stevenson[5], Charles J. R. Williams[6,7], Qiong Zhang[8], Anni Zhao[2], Pascale Braconnot[9], Esther C. Brady[10], Deepak Chandan[11], Roberta D'Agostino[12], Chuncheng Guo[13], Allegra N. LeGrande[14], Gerrit Lohmann[15], Polina A. Morozova[16], Rumi Ohgaito[17], Ryouta O'ishi[18], Bette Otto-Bliesner[10], W. Richard Peltier[11], Xiaoxu Shi[15], Louise Sime[4], Evgeny M. Volodin[19], Zhongshi Zhang[13], and Weipeng Zheng[20]

[1]School of Earth Sciences, University of Melbourne, Parkville, VIC, 3010, Australia

[2]Department of Geography, University College London, London, WC1E 6BT, UK

[3]Department of Atmospheric Sciences, Yonsei University, Seoul, Korea

[4]British Antarctic Survey, High Cross, Madingley Road, CB3 0ET, Cambridge, UK

[5]Bren School of Environmental Sciences and Management, University of California, Santa Barbara, CA, USA

[6]School of Geographical Sciences, University of Bristol, University Road, Bristol BS8 1SS, UK

[7]Department of Meteorology, University of Reading, Earley Gate, P.O. Box 243, Reading RG6 6BB, UK

[8]Department of Physical Geography and Bolin Centre for Climate Research, Stockholm University, 10691, Stockholm, Sweden

[9]Laboratoire des Sciences du Climat et de l'Environnement – IPSL, unite mixte CEA-CNRS-UVSQ, Gif-sur-Yvette, France.

[10]National Center for Atmospheric Research, 1850 Table Mesa Drive, Boulder, CO 80305, USA

[11]Department of Physics, University of Toronto, 60 St. George Street, Toronto, Ontario M5S1A7, Canada

[12]Max-Planck-Institut für Meteorologie, Bundesstrasse 53, 20146, Hamburg, Germany

[13]NORCE Norwegian Research Centre, Bjerknes Centre for Climate Research, Bergen, Norway

[14]NASA Goddard Institute for Space Studies, 2880 Broadway, New York, NY 10025, USA

[15]Alfred Wegener Institute Helmholtz Centre for Polar and Marine Research Bussestr. 24, 27570 Bremerhaven, Germany

[16]Institute of Geography Russian Academy of Sciences, Staromonetny L. 29, Moscow 119017, Russia

[17]Japan Agency for Marine-Earth Science and Technology, 3173-25 Showamachi, Kanazawa-ward, Yokohama, 236-0001, Japan

[18]Atmosphere and Ocean Research Institute, University of Tokyo, Kashiwa, Japan

[19]Marchuk Institute of Numerical Mathematics of the Russian Academy of Sciences, Moscow, Russia

[20]State Key Laboratory of Numerical Modeling for Atmospheric Sciences and Geophysical Fluid Dynamics, Institute of Atmospheric Physics, Chinese Academy of Sciences, Beijing, 100029, China

Correspondence to Josephine R. Brown (josephine.brown@unimelb.edu.au)



**Abstract.** El Niño-Southern Oscillation (ENSO) is the strongest mode of interannual climate variability in the current
climate, influencing ecosystems, agriculture and weather systems across the globe, but future projections of ENSO frequency
and amplitude remain highly uncertain. A comparison of changes in ENSO in a range of past and future climate simulations
can provide insights into the sensitivity of ENSO to changes in the mean state, including changes in the seasonality of
incoming solar radiation, global average temperatures and spatial patterns of sea surface temperatures. As a comprehensive
set of coupled model simulations are now available for both palaeoclimate time-slices (the Last Glacial Maximum, mid-
Holocene and Last Interglacial) and idealised future warming scenarios (one percent per year $CO_2$ increase, abrupt four times
$CO_2$ increase), this allows a detailed evaluation of ENSO changes in this wide range of climates. Such a comparison can
assist in constraining uncertainty in future projections, providing insights into model agreement and the sensitivity of ENSO
to a range of factors. The majority of models simulate a consistent weakening of ENSO activity in the Last Interglacial and
mid-Holocene experiments, and there is an ensemble mean reduction of variability in the western equatorial Pacific in the
Last Glacial Maximum experiments. Changes in global temperature produce a weaker precipitation response to ENSO in the
cold Last Glacial Maximum experiments, and an enhanced precipitation response to ENSO in the warm increased $CO_2$
experiments. No consistent relationship between changes in ENSO amplitude and annual cycle was identified across
experiments.

## 1 Introduction

When the first El Niño-Southern Oscillation (ENSO) event occurred in Earth's history is unclear. However, in light of air-
sea coupled feedbacks, the birth of ENSO must be strongly related to the emergence of the tropical eastern Pacific cold
tongue and its zonal sea surface temperature (SST) contrast with the tropical western Pacific warm pool. It has been
proposed that the gradual uplifting of the Central American continent since around 24 Ma (million years) BP (before present)
triggered the development of the Pacific cold tongue by reducing the surface water exchange between oceans (e.g. Chaisson
and Ravelo, 2000). Palaeo proxies revealed that the Pliocene warm period (~4.5–3.0 Ma), prior to the complete blocking of
Central American Seaway around 2.7 Ma BP (Bacon et al., 2015; O'Dea et al., 2016), recorded a very weak zonal SST
contrast, sometimes referred to as a "permanent El Niño-like state" (Brierley et al., 2009; Fedorov et al., 2013; Fedorov et
al., 2006; Ravelo et al., 2006). Despite the weak mean east-west SST gradient, some proxy records (Scroxton et al., 2011;
Watanabe et al., 2011) and General Circulation Model (GCM) experiments (Burls and Fedorov, 2014; Haywood et al., 2007)
suggest the existence of interannual ENSO variability during the mid-Pliocene, at least sporadically.

Ice sheet dynamics have also been thought to influence the behaviour of ENSO (Liu et al., 2014). Large-scale ice sheets in
the Northern Hemisphere expanded from 2.7 Ma BP onwards (Jansen et al., 2000). Global climate subsequently underwent a
series of glacial-interglacial cycles, with the most recent glacial period reaching maximum levels of global cooling and lower
sea levels around 21~18 ka (thousand years) BP, the so-called "Last Glacial Maximum" (LGM). The tropical climate state
during the LGM was 1–3ºC colder on average than the present day. Reconstructions of LGM ENSO activity are uncertain,



with some studies finding increased ENSO variability (Koutavas and Joanides, 2012; Sadekov et al., 2013) and others finding reduced ENSO variability (Leduc et al., 2009). A recent synthesis of evidence from planktonic foraminifera (Ford et al., 2015) supports reduced ENSO variability but increased seasonality and a deepened equatorial Pacific thermocline during the LGM. Model simulations using an isotope-enabled GCM (Zhu et al., 2017) further assist in reconciling the proxy records, as the model simulates a 30% weakening of ENSO during the LGM but an increased annual cycle contributing to enhanced variability in the foraminifera records of Koutavas and Joanides (2012). Coupled climate models included in the 2$^{nd}$ and 3$^{rd}$ phases of the Paleoclimate Modelling Intercomparison Project (PMIP2 and PMIP3, respectively) simulate a wide range of ENSO changes for the LGM (Masson-Delmotte et al., 2014; Saint-Lu et al., 2015; Zheng et al., 2008).

A number of proxy records provide evidence for a weakening of ENSO variability during the mid-Holocene, although the timing of this weakening varies between records (e.g. Carré et al., 2014; Conroy et al., 2008; Donders et al., 2005; Koutavas and Joanides, 2012; Koutavas et al., 2006; McGregor and Gagan, 2004; McGregor et al., 2013; Rein et al., 2005; Riedinger et al., 2002; Tudhope et al., 2001). On the other hand, Cobb et al. (2013) argue that coral records from the central Pacific do not show a statistically significant reduction in mid-Holocene ENSO variability. Some studies have suggested that disagreement between the magnitude of mid-Holocene ENSO reduction in different proxy records may be due to shifts in the spatial pattern of ENSO variability between the eastern and central Pacific (e.g. Carré et al. 2014; Karamperidou et al., 2015). A synthesis of Holocene ENSO proxy records (Emile-Geay et al., 2016) identifies a sustained reduction in ENSO variability from 3-5 ka, with a reduction of 64% in the central Pacific. During the earlier "mid-Holocene" period from 5.5-7.5 ka, reduced ENSO variance occurs in central, western and eastern Pacific (66%, 50% and 33% respectively) with larger uncertainty ranges (Emile-Geay et al., 2016).

Climate models generally simulate reduced mid-Holocene ENSO activity. For example, transient simulations for part or the whole Holocene period using an intermediate ocean-atmosphere coupled model of the tropical Pacific climate forced by the orbital forcing (Clement et al., 2000), a fully coupled general circulation model with the time-varying climate forcing including orbital, greenhouse gas, meltwater flux, and continental ice sheets (Liu et al., 2014), and a hybrid-type simulation using a combination of the intermediate complexity of earth system model forced orbital forcing and intermediate coupled tropical pacific climate model with varying background state (An et al., 2018), all showed a significant reduction of ENSO intensity during mid-Holocene and its recovery to modern-day ENSO strength around the late Holocene. A study with another set of transient Holocene simulations with coupled climate models confirmed this result, but found that it was the result of chaotic processes (Braconnot et al., 2019).

The mid-Holocene time-slice simulations of PMIP2 and PMIP3, all of which fixed climate forcing at 6,000 years ago (6ka), also showed suppressed ENSO variability in most of the models compared to the pre-industrial perpetual simulations (Braconnot et al., 2007; Chen et al., 2019b; Chiang et al., 2009; Zheng et al., 2008). The reduction of interannual variability in PMIP models was especially dominant over the equatorial central Pacific (An and Bong, 2018; Chen et al., 2019b). However, the reduction of ENSO intensity in the 6ka run of PMIP3 compared to the 0ka run (~ 5% reduction in the standard





deviation of NINO3.4 index from 11 models), in which more state-of-the-art GCMs participated, was rather weaker than that
in PMIP2 (~18% reduction in the standard deviation of NINO3.4 index from 6 models) (An and Choi, 2014; Masson-
Delmotte et al., 2014). A comprehensive model-data comparison (Emile-Geay et al., 2016) found that models
underestimated the reduction in mid-Holocene ENSO variability compared with proxy records, and also simulated an inverse
relationship between the amplitude of the seasonal cycle and ENSO variability which was not evident in proxy
reconstructions.

Over the last millennium, ENSO has exhibited considerable natural variability (Cobb et al., 2003). Multi-proxy
reconstructions of central tropical Pacific SST confirm that vigorous decadal to multi-decadal variability of ENSO occurred
(Emile-Geay et al., 2013). However, as for all the paleoclimate intervals considered, the assessment of ENSO variability
over the last millennium is rather uncertain due to the temporal and spatial sparseness of palaeo-ENSO proxy records (Cobb
et al., 2003; Cobb et al., 2013; Khider et al., 2011). Last Millennium experiments from PMIP3 models showed that ENSO
behaviour may be strongly modulated on decadal- to centennial time scales over the last millennium, and that
teleconnections between ENSO and tropical Pacific climate vary on these time scales (Brown et al., 2016; Lewis and
LeGrande, 2015).

The instrumental records of ENSO for the 20[th] century clearly document the variety of ENSO behaviour, including both
temporal and spatial complexity (Timmermann et al., 2018). ENSO complexity includes its seasonal phase locking (Neelin
et al., 2000; Stein et al., 2011), the interaction with other time-scale climate variability (Eisenman et al., 2005; Levine et al.,
2016; Tang and Yu, 2008; Zhang and Gottschalck, 2002), El Niño-La Niña asymmetry in amplitude, duration, and transition
(An and Jin, 2004; An and Kim, 2018, 2017; Im et al., 2015; Okumura et al., 2011), the diversity in the peak location (i.e.,
Central Pacific- and Eastern Pacific-type El Niño; (Capotondi et al., 2015), and the combination modes due to interaction
between annual and interannual spectra (Stuecker et al., 2015; Timmermann et al., 2018). Interestingly, the dominant
tendency of Eastern Pacific-type El Niño occurrence during 20[th] century was replaced by the Central Pacific-type El Niño
(Ashok et al., 2007; Yeh et al., 2014; Yeh et al., 2009), and all extreme El Niño events (1982-83, 1997-98, and 2015-16)
recorded by the modern instruments occurred around/after the late 20[th] century. The increased frequency of Central Pacific-
type events in recent decades is unusual in the context of a palaeo record for the last 400 years (Freund et al., 2019). Such
distinct changes in ENSO characteristics through 20[th] and 21[st] centuries may be related to low-frequency modulation by
natural variability or the global warming trend due to increasing greenhouse gas concentrations, or a combination of natural
and anthropogenic factors (e.g. An et al., 2008; Cai et al., 2015a; Collins, 2000; Gergis and Fowler, 2009; Timmermann et
al., 1999; Trenberth and Hoar, 1997; Yang et al., 2018; Yeh et al., 2014; Yeh and Kirtman, 2007).

Although it is still a topic of ongoing debate as to whether the future tropical Pacific climate state becomes "El Niño-like" or
"La Niña-like" (referring only to the change in zonal SST gradient) in response to greenhouse warming (An et al., 2012;
Cane et al., 1997; Collins et al., 2010; Lian et al., 2018; Seager et al., 2019), recent multi-model studies of projected changes
in ENSO under global warming suggested no significant change in terms of mean ENSO amplitude compared to the



historical ENSO amplitude (An and Choi, 2015; An et al., 2008; Chen et al., 2017; Christensen et al., 2014; Stevenson, 2012). However, studies have identified robust increases in the extreme hydrological changes associated with El Niño (Cai et al., 2014; Cai et al., 2015a) and La Niña (Cai et al., 2015c), and changes in ENSO-driven precipitation variability (Power et al., 2013). Moreover, even if the global mean temperature is constrained to the limit of 1.5 °C above pre-industrial levels following the Paris Agreement, a doubling of the frequency of extreme El Niño events may occur (Wang et al., 2017).

In this paper, we assess ENSO change through time as simulated in the new generation of coupled atmosphere-ocean climate models for both past and future climates (see Section 2.2). We also compare these new simulations with previous generations of climate models. Detailed comparison of the past climate simulations with proxy records is beyond the scope of the current study, and will be the focus of subsequent research. We consider the change in ENSO amplitude, and its dynamical relationship with the change in the mean climate state under past and future conditions spanning colder past climates (Last Glacial Maximum), past climates with an altered seasonal cycle (Last Interglacial and Mid-Holocene), and idealised warming projections (abrupt four times $CO_2$ and one percent per year $CO_2$) to provide a context for evaluating projections of ENSO change. The methods, models and experiments are introduced in Section 2. Model evaluation is provided in Section 3. In Section 4, the mean state changes in each experiment relative to the pre-industrial control are described. ENSO amplitude changes are presented in Section 5 and changes to ENSO teleconnections are considered in Section 6. In Section 7, the proposed mechanism for the ENSO change through the time are briefly discussed, and conclusions are given in Section 8.

## 2 Methods

This research analyses a total of 140 simulations, across 7 different experiments and 28 climate models. The description of individual simulations is therefore kept brief, and often only the ensemble mean response will be shown. The combined model ensemble will be described in Section 2.1, whilst an overview of the experimental designs are provided in Section 2.2. The common analysis procedure is outlined in Section 2.3.

### 2.1 Models

State-of-the-art coupled global climate models solve the physical equations of the atmosphere and ocean. They are some of the most sophisticated of numerical models, and have been constantly developed for several decades. Globally there are around 40 such models with varying degrees of independence (Knutti et al., 2013). Given the resources required to undertake a single GCM simulation, the international community has settled upon a series of coordinated experiments to facilitate model to model "intercomparison". These are organised under the umbrella of the Coupled Model Intercomparison Project (CMIP). Here we evaluate and analyse simulations from both the previous phase 5 (CMIP5; Taylor et al., 2012), as well as early results from the current phase 6 (CMIP6; Eyring et al., 2016a). Simulations of past climate are included from the Paleoclimate Modelling Intercomparison Project (PMIP), which is part of CMIP. Some of the simulations were carried out



as part of PMIP phase 3 (PMIP3; Braconnot et al., 2012) and other simulations are part of PMIP phase 4 (PMIP4; Kageyama et al., 2018).

For inclusion in this study, a model must have both completed at least one palaeoclimate simulation and provided the required output fields for at least 30 years for both this simulation and the preindustrial control (see section 2.3 for details). The resulting 28 models are listed in Table 1, combined they contain over 35,000 years of monthly ENSO information. Further information about the CMIP5 models is provided in Table 9.A.1 of Flato et al. (2014). The CMIP6 models used in this study are described in more detail in the Supplementary Information and also online on the PMIP4 website https://pmip4.lsce.ipsl.fr/doku.php/database:participants.

## 2.2 Simulations

This study uses simulations consisting of seven different experiments. Four of the experiments are part of the CMIP effort, and form part of the "DECK" set of core simulations (Eyring et al., 2016a): the Pre-industrial and Historical and two idealised future warming scenarios. The study also includes three past climate experiments from the PMIP database, the mid-Holocene and Last Glacial Maximum were included in both PMIP3/CMIP5 and PMIP4/CMIP6, whereas the Last Interglacial was only included in PMIP4/CMIP6.

The mean state and ENSO variability of the models are evaluated in simulations with prescribed historical forcings, known as *historical* simulations (Hoesly et al., 2018; Meinshausen et al., 2017; van Marle et al., 2017). The specification of the *historical* simulation differs slightly between CMIP5 and CMIP6, most notably by the CMIP6 simulations being extended until 2015 C.E. This has minimal influence over the chosen climatological period of 1971-2000. The baseline simulation relative to which all climate changes are calculated is the preindustrial control (*piControl*; Eyring et al., 2016a; Stouffer et al., 2004). The *piControl* simulations represent constant 1850 forcing conditions, and have all reached a quasi-stable equilibrium.

The two idealised warming scenarios (*abrupt4xCO2* and *1pctCO2*) in the CMIP DECK both involve increases in carbon dioxide concentrations. The *abrupt4xCO2* experiment imposes an instantaneous quadrupling of carbon dioxide, to which the coupled climate system is left to equilibrate. The experiment was devised to calculate the climate sensitivity (Gregory et al., 2004). Whilst a full equilibrium of the coupled system will not have occurred within the specified 150 years, there has been substantial progress towards it. The *1pctCO2* experiment is forced with a carbon dioxide increase of one percent per year. This compound increase achieves a quadrupling of carbon dioxide after 140 years, but the climate system is still highly transient. This experiment can be used to calculate the transient climate response (Andrews et al., 2012).

The experimental design for the mid-Holocene (*midHolocene*) and Last Interglacial (*lig127k*) simulations is given by Otto-Bliesner et al. (2017). All *midHolocene* and *lig127k* simulations should have followed this protocol, such that the only significant differences to their corresponding DECK *piControl* simulation are the astronomical parameters and the atmospheric trace greenhouse gas concentrations. In short, astronomical parameters have been prescribed according to



orbital constants from Berger and Loutre (1991) and atmospheric trace greenhouse gas concentrations are based on recent
reconstructions from a number of sources (see section 2.2 in Otto-Bliesner et al. (2017) for details). Note that the different
orbital configurations for *midHolocene* and *lig127k* result in different seasonal and latitudinal distribution of top-of-
atmosphere insolation compared to the DECK *piControl*. For other boundary conditions, these are either small and locally
constrained (e.g. for ice sheets) or there is insufficient spatial coverage to give an informed global estimate (e.g. for
vegetation). These other boundary conditions, including solar activity, palaeogeography, ice sheets, vegetation and aerosol
emissions, were therefore kept as identical to each model's DECK *piControl* simulation. In cases where a boundary
condition can either be prescribed or interactive, such as vegetation, the *midHolocene* and *lig127k* simulations followed the
set-up used in the *piControl* (Otto-Bliesner et al., 2017).

The Last Glacial Maximum (*lgm*) simulation is focused on representing the glacial climate of 21,000 years ago. During this
period, carbon dioxide concentrations dropped by around 100 ppm and large ice-sheets covered the land masses in the
Northern mid to high latitudes. The precise specification of the ice coverage and volume varies between the CMIP5 (Abe-
Ouchi et al., 2015) and CMIP6 (Kageyama et al., 2017) specifications. Whilst this will impact the teleconnections (Jones et
al., 2018), its impact on the tropical Pacific is unclear. The implementation of land-sea changes in regions such as Maritime
Continent is also important (Di Nezio et al., 2016).

### 2.3 Indices and Analysis

This analysis uses a series of standard metrics and measures to describe the simulated ENSO response. These are achieved
using the Climate Variability Diagnostics Package (Phillips et al., 2014), which is part of the ESMValTool (Eyring et al.,
2016b). This software package has previously been used to explore variability in palaeoclimate simulations, although in the
tropical Atlantic rather than Pacific (Brierley and Wainer, 2018). The model output variables required for the analysis are
monthly precipitation rate, monthly surface air temperature and monthly surface temperature. The latter temperature, also
known as skin temperature, is utilised to provide SST on the atmospheric grid (Juckes et al., 2019). Prior to undertaking the
ENSO analysis, the monthly fields of the palaeoclimate simulations are adjusted to represent the changes in the calendar (i.e.
due to changes in the length of months or seasons over time, related to changes in the eccentricity of Earth's orbit and
precession), using the PaleoCalAdjust tool (Bartlein and Shafer, 2019).

The mean state of the present day tropical Pacific is determined using a climatology over the period 1971-2000 for both the
*historical* simulations and observational datasets: the Global Precipitation Climatology Project (GPCP; Adler et al., 2003),
the Hadley Centre Sea Ice and Sea Surface Temperature (HadISST; Rayner et al., 2003) and the 20[th] Century Reanalysis
(C20; Compo et al., 2011)). For the transient *abrupt4xCO2* and *1pctCO2* simulations, the climatology is computed over the
final thirty years. For the other quasi-equilibrium simulations, all available data is considered to create the climatology.
Ensemble mean differences are derived by first calculating the change in climate on each model's grid, and then bilinearly
interpolating onto a common 1° by 1° grid, before averaging across the ensemble members (Brierley and Wainer, 2018).





The state of the tropical Pacific is tracked by the SST anomalies in the NINO3.4 region (5°S-5°N, 120-170°W) (Trenberth, 1997). The anomalies are computed with respect to each simulation's own climatology, with a linear trend removed. All available years are used to assess and composite the NINO3.4 index (even in the transient simulations with a defined climatological period). This choice maximises the number of ENSO events that can be assessed, although it does require the

assumption that changes in the background climatology progress linearly. This assumption is less valid for the *abrupt4xCO2* experiment than the *1pctCO2* experiment, however the ENSO responses show expected coherence across the ensemble, implying the errors introduced are not significant.

The normalised NINO3.4 timeseries are used to composite all years greater than 1 standard deviation to represent El Niño year and all years less than -1 standard deviation to represent La Niña year (Deser et al., 2010). The standard behaviour of

the Climate Variability Diagnostics Package is to use December values of the monthly NINO3.4 time series smoothed with a 3-point binomial filter to identify seasons to composite (Phillips et al., 2014). The process is modified here to allow for changes in the seasonal peak of ENSO activity, potentially associated with the orbital variations. Instead of using smoothed December NINO3.4 values to classify ENSO events, the 3-month smoothed NINO3.4 timeseries is calculated for every month, and the maximum anomaly identified for any month (with a year counted from June to May). This added flexibility

does not quite replicate the standard behaviour over the period 1960-2010, because it additionally identifies 1987 as an El Niño that peaked in August. Previous research has accepted this as a valid El Niño event (e.g. Ramanathan and Collins, 1991).

## 3 Model Evaluation

The ability of the CMIP5/PMIP3 and CMIP6/PMIP4 models (hereafter, "CMIP" models) used in this study to simulate the

present-day SST pattern is first evaluated in comparison with HadISST observations (Rayner et al., 2003) as shown in Figure 1. Consistent with other studies of coupled GCMs (Bellenger et al., 2014; Collins et al., 2010), the models are generally biased toward overly cold SSTs in the central to western equatorial Pacific. Biases toward overly warm SSTs are present in the far eastern Pacific, again a common feature of GCMs generally resulting from a lack of sufficiently deep stratocumulus decks in eastern boundary regions (see review by Ceppi et al., 2017). The northern subtropics also appear to have similar

biases, with colder-than-observed SSTs in the central Pacific and warm biases near the western coast of Mexico. The effect of comparison time period selection is apparent when the *historical* and *piControl* simulations are contrasted (Figure 1c,d versus 1e,f); the *piControl* climate is roughly 1°C colder on average than the *historical*, leading to an apparent exacerbation of the cold-tongue bias and reduction in the warm bias in the eastern equatorial Pacific.

SST biases contribute to errors in the representation of precipitation in the simulations. The Intertropical Convergence Zone

(ITCZ) is generally shifted to the north (Figure 2c,d) leading to a dry bias in the equatorial Pacific, which is particularly pronounced during DJF (Figure 2c). South of the equator, there is a wet bias in the location of the climatological South



Pacific Convergence Zone (SPCZ) (Figure 2c,d) consistent with previously documented tendencies for CMIP-class models to produce a so-called "double ITCZ" (Adam et al., 2018; Zhang et al., 2015). Once again, there are substantial differences between precipitation fields in the *historical* and *piControl* simulations (Figure 2c,d vs. 2e,f), with equatorial precipitation

generally increased in *historical*. This is consistent with the expected intensification of the hydrological cycle under climate change (Held and Soden, 2006; Vecchi and Soden, 2007), for which some observational evidence exists during the 20th century (Durack et al., 2012). These differences are confounded somewhat by the slight differences in the composition of the ensembles for *piControl* and *historical* simulations (see Table 1).

The spatial pattern of ENSO sea surface temperature anomalies is illustrated using the ensemble mean difference between

composite El Niño and La Niña events, shown in Figure 3. The magnitude of simulated *historical* and *piControl* events (Figure 3b,c) is quite close to the observed value (Figure 3a), with peak SST anomaly values of roughly 2.5°C. However, the "centre of action" for ENSO is shifted westward relative to observations; this is a known feature of coupled GCMs, and is related to the biases in mean SST (Bellenger et al., 2014). Because of this westward shift in the peak of El Niño and La Niña events, the magnitude of SST variability is overly weak in the far eastern Pacific (Figure 3b, c). There may be substantial

variation between the individual model simulations, which is documented elsewhere (e.g. Bellenger et al., 2014).

We also evaluate the simulation of global temperature teleconnections with ENSO variability (Figure 4). The observed warming over northern South America, Australia, and much of Southeast Asia during DJF of the El Niño event peak (Figure 4a) is reproduced by the CMIP ensemble mean (Figure 4c), although the magnitudes of the temperature anomalies appear weaker than observed. The teleconnection to the Atlantic and Indian Oceans likewise appears reliable, with comparable

magnitudes of surface warming appearing in the models relative to observations. The models appear to have the most difficulty in representing teleconnections to the higher latitudes; the strong warming over northern North America during DJF (Figure 4a) is significantly underestimated in the models (Figure 4c,e), as is the cooling over northern Eurasia. The same general tendencies hold during JJA (Figure 4b,d,f); here, notable model-observation disagreements are apparent over the eastern half of North America, southern South America, and the southwestern Pacific. This latter feature may relate to

model difficulties with representing SPCZ dynamics (Brown et al., 2013).

Model performance in simulating ENSO temperature teleconnections is reflected in the structure of ENSO precipitation teleconnection biases, shown in Figure 5. In DJF, when El Niño events typically peak, drying occurs in the western Pacific warm pool and over the Amazon in the reanalysis (Figure 5a); this drying persists in JJA but is reduced (Figure 5b). In both cases, the models underestimate the magnitude of South American precipitation teleconnections; additionally, the western

Pacific drying is shifted westward due to the bias in mean SST (Figure 5c,d). Precipitation teleconnections to North America are overly weak in the simulations during both DJF and JJA, as are the tropical Atlantic anomalies.



In summary, the spatial pattern of ENSO SST variability and the remote teleconnections of temperature and precipitation in response to ENSO are reasonably well simulated in the CMIP models and particularly in the ensemble mean. We therefore examine the changes in ENSO in these models under a range of past and future climate conditions.

## 4 Mean State Changes

Changes in the mean state of the tropical Pacific are evaluated for each experiment relative to the *piControl*. This provides the context for consideration of changes in ENSO amplitude and teleconnections in the subsequent Sections. Figures 6 and 7 summarise the seasonal response (DJF and JJA) of the model ensemble to different forcing during the three palaeoclimate experiments (*midHolocene*, *lgm*, and *lig127k*), and two idealised future warming scenarios (*1pctCO2*, and *abrupt4xCO2*) for surface temperature (Figure 6) and precipitation (Figure 7).

For the Last Interglacial (*lig127k*), ensemble changes in surface temperatures (Figure 6e, f) exhibit a strong seasonality that is consistent with *lig127k* minus *piControl* insolation anomalies (see Otto-Bliesner et al., 2017). More specifically, in JJA, regions located at tropical and subtropical latitudes show warming (of about 0.5°C to 2°C). Indeed, during boreal summer, positive insolation anomalies reach their maximum in the Northern Hemisphere (NH) and extend into the tropics and the Southern Hemisphere (SH). In contrast, in DJF, negative insolation anomalies are large in SH and NH equatorward of 40°N, and tropical and subtropical latitudes show cooling (mostly of about 1°C). Similar patterns, although much weaker and spatially constrained, are shown for the *midHolocene* simulation (Figure 6a and Figure 6b for DJF and JJA, respectively), with small anomalies in both seasons. A small cooling of ~0.5°C is shown over the equatorial regions of the eastern (central) Pacific during DJF (JJA). No warming, however, is shown in the *midHolocene*, in either season. Based on the sign and magnitude of the insolation anomalies for both the *midHolocene* and (more so) *lig127k*, the model ensemble mean state changes for both of these simulations are consistent with previous modelling and proxy reconstruction studies (see e.g. Otto-Bliesner et al., 2013).[1]

The *lgm* ensemble mean shows cooling of around 2–3°C in the tropical Pacific in both DJF and JJA (Figure 6c,d), consistent with previous modelling studies and proxy reconstructions (Ballantyne et al., 2005; MARGO Project Members et al., 2009; Masson-Delmotte et al., 2014; Otto-Bliesner et al., 2009). For the idealised future scenarios, the tropical Pacific warms (by 1-3°C in the *1pctCO2* case and more than 3°C in the *abrupt4xCO2* case), with largest warming in the equatorial region. In the case of the *abrupt4xCO2* simulations, the ensemble mean warming is largest in the eastern and central Pacific, particularly in DJF; whereas the *1pctCO2* ensemble shows enhanced warming extending across the equatorial Pacific. This enhanced equatorial warming is a recognised feature of anthropogenic climate warming (DiNezio et al., 2009; Liu et al.,

---

[1] Further analysis and description of the results of these PMIP4 experiments can be found in other articles in this special issue – e.g. Brierley et al. for *midHolocene* and Otto-Bliesner et al. for *lig127k*.



2005; Xie et al., 2010) and has important implications for ENSO, leading to more frequent extreme El Niño events in a warmer climate (Cai et al., 2014; Wang et al., 2017).

Regarding ensemble precipitation anomalies, in the *lig127k* simulations (Figure 7e,f), the weaker Australian monsoon (drier conditions over northern Australia in DJF) and the enhanced North America monsoon (wetter conditions over northern South America in JJA) are consistent with a northward shift of the mean seasonal position of the ITCZ (over the oceans) and the
associated tropical rainfall belt (over the continents). As for surface temperatures, the *midHolocene* ensemble mean (Figure 7a,b) shows similar spatial patterns of mean state precipitation change to the *lig127k* case, but with smaller magnitude. The Australian monsoon (North American monsoon) is weaker (stronger) in the *midHolocene* ensemble mean relative to *piControl*, just less so than the *lig127k*. Drying in the *midHolocene* is larger in the western Pacific than for *lig127k*. Mechanisms and drivers for precipitation changes over the tropics in Last Interglacial climates are still unclear and represent
an active area of research (see e.g. Scussolini et al., 2019; Otto-Bliesner et al. *in prep.*).

Precipitation changes in the *lgm* ensemble (Figure 7 c,d) show a drying over the Maritime Continent, Australia and Southeast Asia. Precipitation in the ITCZ over the tropical Pacific is also reduced, particularly in JJA, in response to cooler SSTs. Precipitation is increased in the western Pacific and on the northern edge of the SPCZ, indicating a northward displacement of the SPCZ as found in previous studies (Saint-Lu et al., 2015). In the *1pctCO2* and *abrupt4xCO2* simulations (Figure 7 g-
j), precipitation increases in the equatorial Pacific where SST warming is greatest (e.g. Chadwick et al., 2013; Xie et al., 2010), with some drying on the northern edge of the ITCZ, particularly in the eastern Pacific. Drying also occurs in the southeast Pacific, where warming is relatively small and trade winds are intensified, leading to drying of the eastern edge of the SPCZ in DJF (Brown et al., 2013; Widlansky et al., 2013).

## 5 ENSO Amplitude Changes

The amplitude of ENSO, as measured by the standard deviation of SST from the NINO3.4 region, is shown for each model in each experiment in Figure 8. The amplitudes of ENSO in the *piControl* simulation from each model are also shown for reference. The percentage change in ENSO amplitude in the experiments relative to *piControl* is shown in Figure 9.

In the *midHolocene* simulations, a large majority (24 out of 27) of models show a decrease in ENSO variability, with the only exceptions being CSIRO-Mk3-6-0, INM-CM4-8 and MRI-CGCM3. Most of these changes are small, however, with
few models showing more than a ~20% decrease (Figures 8a and 9a). This is consistent with previous model studies, which generally show a smaller reduction in *midHolocene* ENSO amplitude than implied by proxy records (e.g. Emile-Geay et al., 2016), as discussed in Section 1. A similarly consistent reduction in ENSO amplitude is found for the *lig127k* simulations (Figure 8c and 9c), with 9 out of 11 models showing a reduction in amplitude, typically of at least 20%.

In contrast, a much less consistent response is found for the *lgm*, *1pctCO2* and *abrupt4xCO2* simulations. In all of these
simulations, the sign of change in amplitude of ENSO is approximately equally spread between increases and decreases



across the set of models. In the *lgm* simulations (Figure 8b and 9b), some models (e.g. FGOALS-s2) show a large decrease in variability of over 40% and others (e.g. IPSL-CM5A-LR) show large increases of up to ~40%. Likewise, in the *1pctCO2* simulations (Figure 8d and 9d), ENSO variability is again highly model-dependent, with the range including large decreases of over ~20% in some models (e.g. CCSM4) to large increases of up to ~40% in others (e.g. MPI-ESM-P). The same is true

for the *abrupt4xCO2* simulations (Figure 8e and 9e), with the range including large decreases of over ~40% in some models (e.g. GISS-E2-R) to large increases of up to ~50% in others (e.g. CSIRO-Mk3-6-0). This is consistent with previous studies showing little agreement on future projections of ENSO amplitude change (Collins et al., 2010; Collins et al., 2014).

Comparing the absolute magnitudes of ENSO amplitude in all simulations (Figure 8), the standard deviation ranges (i.e. between the models showing the smallest and largest standard deviations) are ~0.7°C in the *midHolocene*, ~1.1°C in the *lgm*,

~0.2°C in the *lig127k*, ~0.7°C in the *1pctCO2* simulation and ~1°C in the *abrupt4xCO2* simulation. The cold climate simulation and the extreme future run therefore show the largest spread between models, suggesting a lack of model agreement, whereas the *mid-Holocene* and *lig127k* simulations as well as the gradual future run have a much smaller spread between models.

Comparing the change in ENSO amplitude in all simulations (Figure 9), we find that the *midHolocene* and *lig127k*

simulations have high inter-model agreement on the sign of the response, consistently showing lower ENSO variability relative to the *piControl* in both cases. The common factor between these simulations is the change in seasonality of insolation, which in both cases is increased in boreal summer, leading to a damped ENSO via a range of mechanisms discussed in Section 1 and also Section 7. In contrast, there is much less inter-model agreement in the cold climate simulation (i.e. the *lgm*) and the gradual and extreme future warming runs (i.e. the *1pctCO2* and *abrupt4xCO2* simulations,

respectively).

## 6 ENSO Patterns and Teleconnections

The anomalous pattern of El Niño minus La Niña SST composite for each experiment relative to *piControl* (see Figure 3c) is shown in Figure 10. The *midHolocene* and *lig127k* patterns (Figure 10a, c) show negative SST anomalies in the central equatorial Pacific, indicating weakening of event amplitude, consistent with the average weakening of ENSO variability in

these experiments (see Section 5 and Figure 9). There is a much larger weakening of SST variability in the *lig127k* than the *midHolocene* case. The *lgm* SST pattern (Figure 10b) shows negative anomalies in the central to western Pacific, indicating either an eastward shift of the ENSO pattern and/or weaker central Pacific variability. On the other hand, both the *1pctCO2* and *abrupt4xCO2* composites (Figure 10d, e) show positive Pacific SST anomalies associated with ENSO at the equator, with the largest values in the central Pacific, and high model agreement. This suggests an increased ENSO variability among

the ensemble, particularly for the *abrupt4xCO2* simulations, despite the disagreement between model ENSO amplitude changes discussed in Section 5.





The global temperature and precipitation teleconnections with ENSO for each experiment relative to *piControl* are shown in Figures 11 and 12. As discussed above, both *lig127k* and *midHolocene* simulations show weaker ENSO SST variability relative to *piControl* (Figure 10a, c). The *lig127k* simulations have a much greater weakening of the ENSO SST and
temperature patterns (Figure 10c and 11e, f) than any of the other simulations (although based on a small number of models), with cooler SSTs in the central Pacific. The *midHolocene* ENSO SST and temperature pattern (Figure 10a and 11a, b) is a weaker version of the *lig127k* response. The ENSO precipitation teleconnection in the *lig127k* simulations (Figure 12e, f) consists of a weakening of the *piControl* ENSO precipitation pattern, with much drier conditions in the equatorial Pacific and over the SPCZ during El Niño events. The *midHolocene* ENSO precipitation pattern (Figure 12a, b) is again a weaker
version of the *lig127k* ENSO precipitation response.

The *lgm* simulations show cooler SSTs in the western tropical Pacific in the ENSO composite (Figure 10b), consistent with a weakening of ENSO variability in this region. Cool anomalies over Australia and warm anomalies over North America are also evident (Figure 11c, d). As expected, given the colder global mean temperatures (Figure 6), precipitation associated with ENSO is reduced in the tropical Pacific and the overall hydrological cycle is weaker (Figure 12 c, d). Remote responses
include wetter conditions over Australia and drier conditions over North America, somewhat resembling a La Niña pattern, although with low model agreement.

In contrast to the *lgm* experiments, the *1pctCO2* and *abrupt4xCO2* simulations are much warmer than the other experiments (Figure 6). The largest temperature anomalies are seen for the *abrupt4xCO2* simulations (Figure 11i, j), which also had increased amplitude of SST variability in the central Pacific (Figure 10e). The *abrupt4xCO2* experiment shows warmer
temperatures globally during El Niño events, particularly over continents and high northern latitudes (except Greenland in DJF). It is possible that elements of the pattern in this experiment arise from the linear detrending failing to sufficiently remove the transient changes in mean state (Section 2.3). The *1pctCO2* simulations (Figure 11g, h) show a similar but much weaker response. The precipitation response to ENSO (Figure 12) is enhanced in the *abrupt4xCO2* and *1pctCO2* simulations relative to *piControl*, with increases in precipitation in the equatorial Pacific and decreases in the subtropics, as ENSO
influences tropical atmospheric circulation and therefore the hydrological cycle (Lu et al., 2008; Nguyen et al., 2013). This is consistent with previous studies showing intensified ENSO temperature and precipitation impacts in a warmer climate (e.g. Power et al., 2013; Power and Delage, 2018).

The underlying relationships between aspects of the mean state and ENSO are investigated for all models and experiments (Figure 13). The change in ENSO amplitude (relative to *piControl*) is plotted against the change in annual cycle in Figure
13a, with no significant correlation between the two variables. The change in ENSO amplitude was also found to have no significant correlation with the change in zonal SST gradient (defined as 5° S–5° N, 150° E–170° W minus 5° S–5° N, 120° W–90° W), shown in Figure 13b. Finally, the relationship between extreme El Nino and the mean state (Cai et al., 2014) is investigated (Figure 13c). The mean state is represented as changes in meridional SST gradient in the eastern Pacific – defined as the average SST over the off-equatorial region (5° N–10° N, 150° W–90° W) minus the average over the
equatorial region (2.5° S–2.5° N, 150° W–90° W). The strength of El Nino rainfall is represented as the changes in ENSO





composite precipitation over the NINO3 region (5° S–5° N, 150° W–90° W) normalised by the NINO3.4 standard deviation used to identify the composited events. This normalisation aims to remove the impact of the changes in ENSO variability documented between the experiments (e.g. Power and Delage, 2018). The strong negatively relationship is consistent with the analysis demonstrated by Cai et al. (2014) and (Collins et al., 2019), but this analysis approach allows to that relationship

to be visualised across many more simulations and experiments. This relationship appears to be fundamental feature of ENSO behaviour, rather than just a response to greenhouse gas forcing.

## 7 Mechanisms and Discussion

The mean state of the tropical Pacific during the Pliocene warm period featured sustained El Niño-like conditions (Fedorov et al., 2006; Wara et al., 2005). Such weak zonal gradient in mean equatorial Pacific SST and the deep eastern Pacific

thermocline are linked to strongly suppressed trade winds. The suppressed trade winds, or westerly so-called 'super-rotation flow', can be driven by the equatorward westerly eddy momentum flux originating from the enhanced poleward Rossby wave pumping due to the stronger tropical convection under the warm ocean surface (Arnold et al., 2012; Tziperman and Farrell, 2009). Whilst individual El Niño events occurred during the Pliocene warm period (Ford and Ravelo, 2019), it is likely that the weak zonal contrast in mean SST (i.e., El Niño-like mean condition) was less favourable for ENSO occurrence

(Brierley, 2013; Manucharyan and Fedorov, 2014). The mid-Holocene, another ENSO suppression period, featured a stronger zonal gradient in the tropical Pacific mean SST than the 20th century, namely "La Niña-like conditions" (Barr et al., 2019; Gagan and Thompson, 2004; Koutavas et al., 2002; Luan et al., 2012; Shin et al., 2006). These contradictory responses imply that the dynamical mechanisms determining the relationship between the zonal gradient in mean SST and ENSO amplitude (e.g., Sadekov et al., 2013) must consist of several processes. This lack of consistent relationship between annual

cycle between the Pliocene and mid-Holocene is supported by the results presented here, shown in Fig. 13b.

During the mid-Holocene, the reduced tropical insolation led to the cooling of the tropical Pacific, directly producing a La Niña-like condition. Under La Niña-like conditions, the air-sea coupling strength is reduced due to a suppressed convective instability, and thus ENSO variability is suppressed (Liu et al., 2000; Roberts et al., 2014). The stronger seasonality in insolation over the Northern Hemisphere associated with the precession cycle resulted in a stronger annual cycle, which

could also act to reduce ENSO variability through the intensified annual-frequency entrainment (Liu, 2002; Pan et al., 2005). A similar but stronger precession effect due to the higher eccentricity during the last interglacial period was also found to have a relatively weak ENSO amplitude in palaeo-proxy records (Hughen et al., 1999; Tudhope et al., 2001) and climate model simulations (An et al., 2017; Salau et al., 2012). However, the mid-Holocene simulations of PMIP2/3 mostly showed a reduction of both annual cycle and ENSO amplitude (An and Choi, 2014; Masson-Delmotte et al., 2014).

The reduced annual cycle over the tropical eastern Pacific is attributed to the relaxation of eastern Pacific upper-ocean stratification due to the annual downwelling Kelvin wave forced by western Pacific wind anomalies (Karamperidou et al.,



2015) or the deepening of ocean mixed layer depth associated with the northward shift of the ITCZ (An and Choi, 2014). Therefore, the mid-Holocene ENSO variability in PMIP2/3 may be deemed to the result of the counterbalance between the reduction due to the weaker air-sea coupling and the intensification due to the reduced frequency entrainment (An and Choi, 445 2014). Other factors may include coupling of the circulation in the eastern Pacific with the North American monsoon (implying dynamical damping of upwelling in the eastern Pacific), and an increased southeast Asian monsoon which strengthens winds in the western Pacific. Alternatively, An et al. (2010) and An and Choi (2013) argue that the change in annual cycle amplitude is not a cause of change in ENSO amplitude, it is the changes in the mean climate state that modify both the ENSO and annual cycle amplitudes in the opposite way. The analysis presented here would appear to support this 450 argument there is no consistent relationship between changes in the amplitude of the annual cycle and changes in the ENSO variability (Fig. 13a).

ENSO variability can be suppressed or enhanced by remote forcing. For example, the enhanced Asian summer monsoon also leads to La Niña-like conditions via increasing strength of the tropical Pacific trade winds and the resultant enhanced equatorial upwelling (Liu et al., 2000). Sensitivity experiments with fully coupled climate models demonstrate that greening 455 of the Sahara during the mid-Holocene could reduce ENSO variability through affecting the Atlantic Niño (Zebiak, 1993) and Walker Circulation, finally decreasing upwelling and deepening of the thermocline in the eastern Pacific (Pausata et al., 2017). The freshwater perturbation experiments, so-called "water hosing experiment" that lead to a weakening of the Atlantic Ocean meridional overturning circulation, showed a reduced seasonal cycle and enhanced ENSO variability through the inter-basin atmospheric teleconnection (Braconnot et al., 2012; Masson-Delmotte et al., 2014; Timmermann et al., 2007).

More sophisticated feedback analysis revealed that the reduction of ENSO variability is due to either the increase of the negative feedback by the mean current thermal advection (An and Bong, 2018) or the reduction of the major positive feedback processes (thermocline, zonal advection and Ekman feedbacks) (Chen et al., 2019a; Tian et al., 2017). The negative feedback due to the thermal advection by the mean current was intensified by the stronger cross-equatorial winds associated with the northward migration of the ITCZ (e.g., An and Choi, 2014), and the positive dynamical feedback was suppressed 465 due to the strengthening of the mean Pacific subtropical cell (Chen et al., 2019a). Therefore, the linear stability of ENSO during the mid-Holocene was reduced through the dedicated balance among the various feedback processes, but the change in each feedback process is model-dependent. External processes were also proposed as a suppression mechanism for mid-Holocene ENSO. For example, the Pacific meridional mode became weaker during the mid-Holocene, thereby ENSO has relatively less chance to be triggered (Chiang et al., 2009); the weaker ocean stratification due to the warm water subduction 470 from the subtropical ocean decreases ENSO stability (Liu et al., 2000).

How ENSO activity will change in response to anthropogenic global warming still remains uncertain (Cai et al., 2015b; Christensen et al., 2014; Collins et al., 2010). Observations show that ENSO variability has increased under greenhouse warming in the recent past (Zhang et al., 2008), which is also shown in CMIP5 climate model simulations (Cai et al., 2018). During the transient period of global warming, the tropical SSTs warm much faster than the subsurface ocean and leads to a



shallower and stronger thermocline in the equatorial Pacific (An et al., 2008), which enhances the ocean-atmosphere coupling and amplifies the ENSO variability (Zhang et al., 2008). A gradually intensified ENSO from mid- to late Holocene also appears in a long-transient simulation since the last 21,000 years (Liu et al., 2014) and of the last 6000 years (Braconnot et al., 2019). For the equilibrium response to global warming, the subsurface ocean will eventually warm up and reduce the vertical temperature gradient and weaken the ENSO variability. For instance, during the Pliocene warm period, the most

recent period in the past with carbon dioxide concentrations similar or higher than today, SST reconstructions from the tropical Pacific show a reduced zonal SST gradient during this period, implying sustained El Niño-like conditions (Dekens et al., 2007; Fedorov et al., 2006; Wara et al., 2005).

The tropical climate mean state response to current global warming is still a topic of debate as to whether it will be El Niño-like or La Niña-like (Cane et al., 1997; Collins, 2005); Merryfield 2006; (An et al., 2012; Collins et al., 2010; Lian et al.,

2018). Moreover, the global warming-induced tropical Pacific SST pattern seems to be less effective in changing ENSO activity (An and Choi, 2015). The strong internal modulation of ENSO activity over decadal-to-centennial timescales also obscures the actual global warming impact on ENSO variability (e.g. Wittenberg, 2009). So far, the future ENSO activity reflected in SST anomalies obtained from CMIP5 models is not distinguishable from the historical ENSO activity (Christensen et al., 2014).

The enhancement and increasingly frequent occurrence of ENSO-driven extreme atmospheric responses to future global warming are strongly supported by model studies. This include extreme rainfall events and extreme equatorward swings of the SPCZ (Cai et al., 2014; Cai et al., 2015a; Cai et al., 2012) and extreme weather events through teleconnections (Cai et al., 2015b; Yeh et al., 2018). The changes in these extremes are due to the nonlinearity of atmospheric response to ENSO SSTs, especially with a warmer ocean surface. The changing amplitude of the extremes with changing meridional SST gradient is a

feature of past climates as well as future climates. This is demonstrated by Fig. 13c, which shows an increase in the ENSO composited precipitation with increased SST gradient. However, most current GCMs still inaccurately simulate many aspects of the historical ENSO such as the far westward extent of the Pacific cold tongue (Taschetto et al., 2014), ENSO-related precipitation anomalies (Dai and Arkin, 2017), ENSO feedback (Bellenger et al., 2014; Kim and Jin, 2011; Lloyd et al., 2012), and ENSO asymmetry in amplitude, duration, and transition (e.g. Chen et al., 2017; Zhang and Sun, 2014).

Therefore, the accuracy of the future projections of ENSO will be guaranteed only when most coupled GCMs can simulate the observed modern-day ENSO more skilfully than at present.

## 8 Summary and Conclusions

We have presented a summary of ENSO amplitude and teleconnections changes in the most recent previous generation (CMIP5) and the new generation (CMIP6) of coupled climate models for past and future climates. The analysed simulations





include the last glacial maximum climate (*lgm*), past interglacial climates (*lig127k* and *midHolocene*), and future idealised
      projections (*abrupt4xCO2* and *1pctCO2*), using the pre-industrial climate (*piControl*) as the reference state.

We first evaluated a 30-year climatology from *historical* simulation against HadISST observation from 1971-2000. Similarly
to the previous generations of climate models, the CMIP5 and CMIP6 models have cold biases in SST in the central to
western equatorial Pacific, as well as in the subtropical Pacific. Warm SST biases are present in the eastern equatorial
Pacific. The *piControl* climate is about 1°C colder on average than the *historical* climate, leading to an apparent exacerbation
      of the cold-tongue bias and reduction in the warm bias in the eastern equatorial Pacific. These biases in SST lead to a strong
      and northward shift of ITCZ, a dry bias along the equatorial Pacific, and appearance of a "double ITCZ" as in previous
      CMIP-class simulations. The simulated ENSO pattern well resembles the observations with a slight displacement to the
      west, similarly the ENSO temperature and precipitation teleconnections are well simulated compared with observations.

The mean state changes were examined relative to the *piControl*. In the *lig127k* simulations, strong seasonal insolation
      anomalies lead to tropical and subtropical SST cooling of 0.5-2°C in DJF and JJA. No large-scale warming is found in the
      *midHolocene* simulations, but a slight cooling of 0.5°C occurs in the eastern Pacific in DJF and in the central Pacific in JJA.
      In the *lgm* simulations, 2-3°C cooling is found in the tropical Pacific. For the future scenarios, the tropical Pacific warms by
      1-3°C in the *1pctCO2* case and more than 3°C in the *abrupt4xCO2* case. During *lig127k* and *midHolocene* simulations, the
ITCZ shifts northward, leads to a weakened Australian summer monsoon and enhanced North American summer monsoon.
      In the *lgm* simulations, the ITCZ is intensified over the tropical Pacific, with drier conditions over the Maritime Continent,
      Australia and Southeast Asia, while wetter conditions are found in the western Pacific and on the northern edge of the SPCZ.
      In the *1pctCO2* and *abrupt4xCO2* simulations, precipitation increases in the equatorial Pacific following the largest SST
      warming, with some drying on the northern edge of the ITCZ and in southeast Pacific in DJF.

The majority of models show a decrease in ENSO variability in the *lig127k* and *midHolocene* simulations. The reduction of
      ENSO variability in *lig127k* ranges to more than 40%, while only one model shows more than a ~20% decrease in ENSO
      variability in *midHolocene*. This is consistent with previous model studies of mid-Holocene ENSO, which generally show a
      smaller reduction in amplitude than implied by proxy records.

The changes in ENSO variability in *lgm*, *1pctCO2* and *abrupt4xCO2* simulations are highly model dependent, with the sign
of change approximately equally divided between increases and decreases of up to 40% to 50% across the set of models.
      This is also consistent with previous studies showing little agreement on ENSO amplitude change under LGM conditions
      (Masson-Delmotte et al., 2014) or in future projections (Collins et al., 2014).

The ensemble mean weakening of ENSO in the *lgm* simulations is characterised by a cooling in the central and western
Pacific. Cooling over Australia and warming over North America are also evident. The changes in the ENSO temperature
teleconnections show cooling in the central Pacific in *lig127k* and *midHolocene*, and significant global warming in the
      *abrupt4xCO2* simulation, with strong warming amplification over continents and high northern latitudes. The *1pctCO2*





simulations show a similar warming pattern but with much weaker magnitude. Precipitation teleconnections follow the change in ensemble mean ENSO amplitude, with a weakening of the climatological *piControl* ENSO precipitation response in *lgm*, *lig127k* and *midHolocene* cases. In the *lgm* simulations, a weaker hydrological cycle due to cooler temperatures may 540 also play a role. The precipitation response to ENSO is significant in the *abrupt4xCO2* and *1pctCO2* simulations, with an amplification of the climatological ENSO teleconnection pattern. This is consistent with previous studies showing increased ENSO precipitation responses in a warmer climate.

This study has provided an overview of changes in the mean state and ENSO in a set of past and future climate simulations from PMIP3/CMIP5 and PMIP4/CMIP6 models. We have not provided a comprehensive analysis of any aspect of these 545 changes, and the set of CMIP6 models included is necessarily incomplete. Future work will focus on deepening understanding of the complex mechanisms driving interactions between changes in the mean state and ENSO. A more detailed comparison with palaeo-ENSO proxy records will also be required to evaluate model simulations of ENSO in past climates (e.g. Lu et al., 2018). Improved understanding of changes in ENSO in past climates can contribute to model evaluation, understanding of ENSO dynamics and constraining projections of future change.


**Code and data availability**

The CMIP5 and CMIP6 model data used in this study are available from the Earth System Grid Federation. All the scripts used for the analysis and to produce the figures in this study are provided on the GitHub repository: https://github.com/chrisbrierley/PMIP4-enso. All the ENSO diagnostics are also available via the PMIP variability database 555 at past2future.org.

**Author contributions**

J.R.B. coordinated the research. C.M.B. performed the simulation analysis. J.R.B, C.M.B., S.-I.A., M.-V.G.,S.S., C.J.R.W., Q.Z. and A.Z. authored the manuscript with edits from R.D. Other authors contributed simulations towards the analysis.

**Competing interests**

The authors declare no competing interests.

**Acknowledgements**

J.R.B acknowledges support from the ARC Centre of Excellence for Climate Extremes (CE170100023). C.M.B., CJ.R.W., P.B and R.D. acknowledge the JPI-Belmont-funded PACMEDY programme (NE/P006752/1 and ANR-15-JCLI-0003-01). S.-I.A. was supported by the National Research Foundation of Korea (NRF) grant funded by the Korea government (MSIT) 565 (NRF-2018R1A5A1024958). C.M.B. was also funded in part by NERC (NE/S009736/1). M.V.G. acknowledges support



from NERC research grant NE/P013279/1. Q.Z. acknowledges support from the Swedish Research Council VR projects 2013-06476 and 2017-04232. Ru.O. acknowledges support from the Integrated Research Program for Advancing Climate Models (TOUGOU programme) from the Ministry of Education, Culture, Sports, Science and Technology (MEXT), Japan. The simulations using MIROC-ES2L were conducted on the Earth Simulator of JAMSTEC. We acknowledge the World

Climate Research Programme, which, through its Working Group on Coupled Modelling, coordinated and promoted CMIP6. We thank the climate modeling groups for producing and making available their model output, the Earth System Grid Federation (ESGF) for archiving the data and providing access, and the multiple funding agencies who support CMIP6 and ESGF. PMIP is endorsed by both WCRP/WGCM and Future Earth/PAGES. This research arose out of a workshop hosted at University College London by the PMIP working group on Past2Future: insights from a constantly varying past.

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



**Table 1:** List of models included in study and length of simulations based on number of years of data available for NINO3.4
in the CVDP archive. Additional information about CMIP6/PMIP4 models (indicated in bold) is provided in the
Supplementary Material.

| Model | CMIP gen. | Pi-Control | historical | mid-Holocene | lgm | lig127k | 1pctCO2 | abrupt4xCO2 |
|---|---|---|---|---|---|---|---|---|
| AWI-ESM-1-1-LR | **CMIP6** | 100 | - | 100 | 100 | 100 | - | - |
| BCC-CSM1-1 | CMIP5 | 500 | 163 | 100 | - | - | 140 | 150 |
| CCSM4 | CMIP5 | 1051 | 156 | 301 | 101 | - | 156 | 151 |
| CESM2 | **CMIP6** | 1200 | 165 | 700 | - | 700 | 150 | 150 |
| CNRM-CM5 | CMIP5 | 850 | 156 | 200 | 200 | - | 140 | 150 |
| COSMOS-ASO | CMIP5 | 400 | - | - | 600 | - | - | - |
| CSIRO-Mk3-6-0 | CMIP5 | 500 | 156 | 100 | - | - | 140 | 150 |
| CSIRO-Mk3L-1-2 | CMIP5 | 1000 | 150 | 500 | - | - | 140 | - |
| FGOALS-f3-L | **CMIP6** | 561 | 165 | 500 | - | 500 | 160 | 160 |
| FGOALS-g2 | CMIP5 | 700 | 115 | 680 | 100 | - | 244 | 258 |
| FGOALS-g3 | **CMIP6** | 700 | - | 500 | - | 500 | - | - |
| FGOALS-s2 | CMIP5 | 501 | - | 100 | - | - | 140 | 150 |
| GISS-E2-1-G | **CMIP6** | 851 | 165 | 100 | - | 100 | 51 | 151 |
| GISS-E2-R | CMIP5 | 500 | 156 | 100 | 100 | - | 151 | 151 |
| HadGEM2-CC | CMIP5 | 240 | 145 | 35 | - | - | - | - |
| HadGEM2-ES | CMIP5 | 336 | 145 | 101 | - | - | 140 | 151 |
| HadGEM3-GC31 | **CMIP6** | 100 | - | 100 | - | 200 | - | - |
| INM-CM4-8 | **CMIP6** | 531 | 165 | 200 | 200 | 100 | 150 | 150 |
| IPSL-CM5A-LR | CMIP5 | 1000 | 156 | 500 | 200 | - | 140 | 260 |
| IPSL-CM6A-LR | **CMIP6** | 1200 | 165 | 550 | - | 550 | 150 | 900 |
| MIROC-ES2L | **CMIP6** | 500 | 165 | 100 | 100 | 100 | 150 | 150 |
| MIROC-ESM | CMIP5 | 630 | 156 | 100 | 100 | - | 140 | 150 |
| MPI-ESM-P | CMIP5 | 1156 | 156 | 100 | 100 | - | 140 | 150 |
| MRI-CGCM3 | CMIP5 | 500 | 156 | 100 | 100 | - | 140 | 150 |
| MRI-ESM2-0 | **CMIP6** | 701 | 165 | 200 | - | - | 151 | 151 |
| NESM3 | **CMIP6** | 100 | 165 | 100 | - | 100 | 150 | 150 |
| NorESM1-F | **CMIP6** | 200 | - | 200 | - | 200 | - | - |
| UofT-CCSM-4 | **CMIP6** | 100 | - | 100 | - | - | - | - |

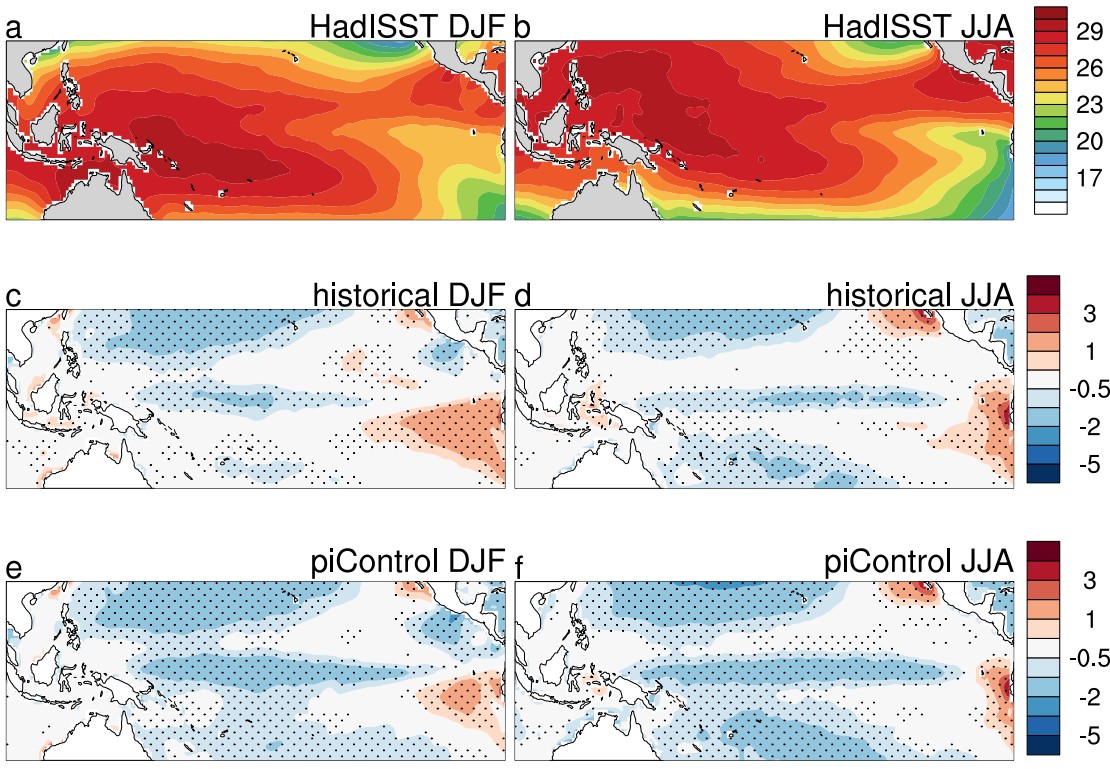

**Figure 1**: Ability of the ensemble to simulate present-day sea surface temperature (SST) patterns: (a) DJF and (b) JJA SST climatology from HadISST observational dataset (Rayner et al., 2003) between 1971-2000, (c) DJF and (d) JJA model ensemble mean SST in historical simulations minus HadISST observations between 1971-2000, and (e) DJF and (f) JJA model ensemble mean SST in pre-industrial control simulations minus HadISST observations. Units are °C.





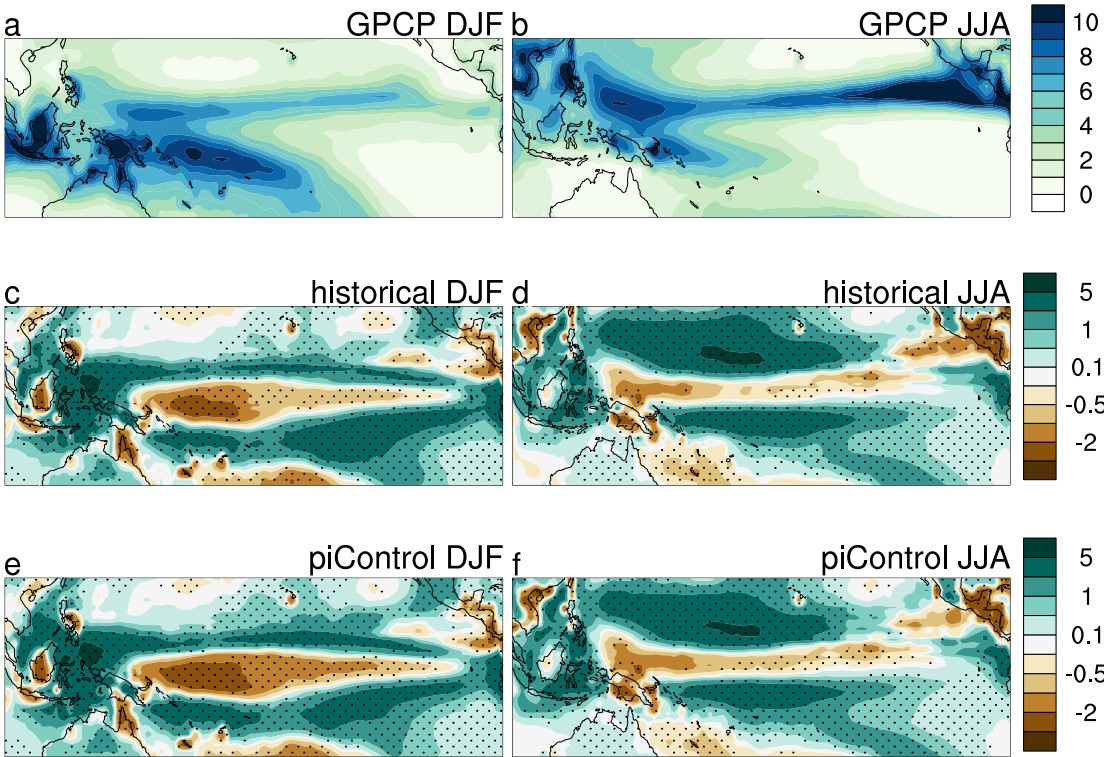

**Figure 2**: Ability of the ensemble to simulate present-day precipitation patterns: (a) DJF and (b) JJA SST climatology from GPCP observational dataset (Adler et al., 2003) between 1979-1999, (c) DJF and (d) JJA model ensemble mean precipitation in historical simulations minus GPCP observations between 1979-1999, and (e) DJF and (f) JJA model ensemble mean precipitation in pre-industrial control simulations minus GPCP observations. Units are mm/day.



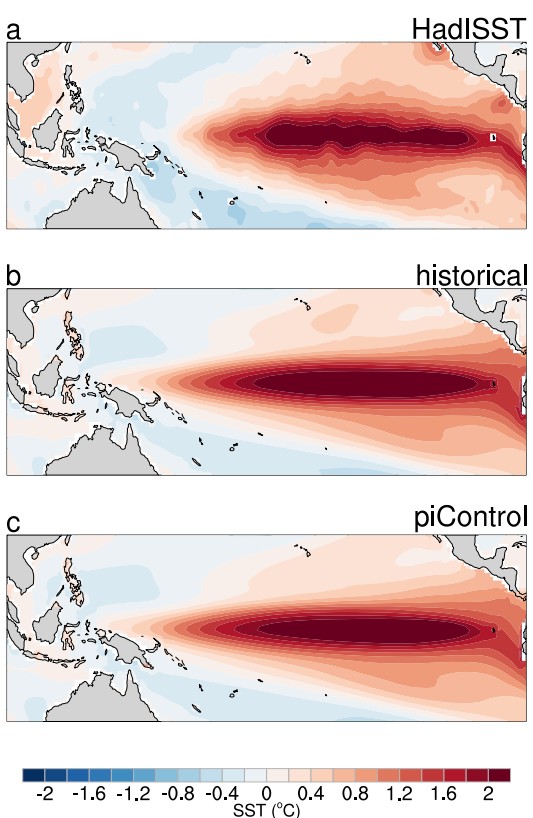

**Figure 3:** Evaluation of the ENSO SST anomaly pattern. Composite El Niño minus La Niña sea surface temperature anomaly from (a) HadISST observational dataset between 1871-2012, (b) model ensemble average from the historical simulations between 1850-2005 (for CMIP5 models) or 1850-2015 (for CMIP6 models), and (c) model ensemble average from the pre-industrial control simulations. Units are °C.





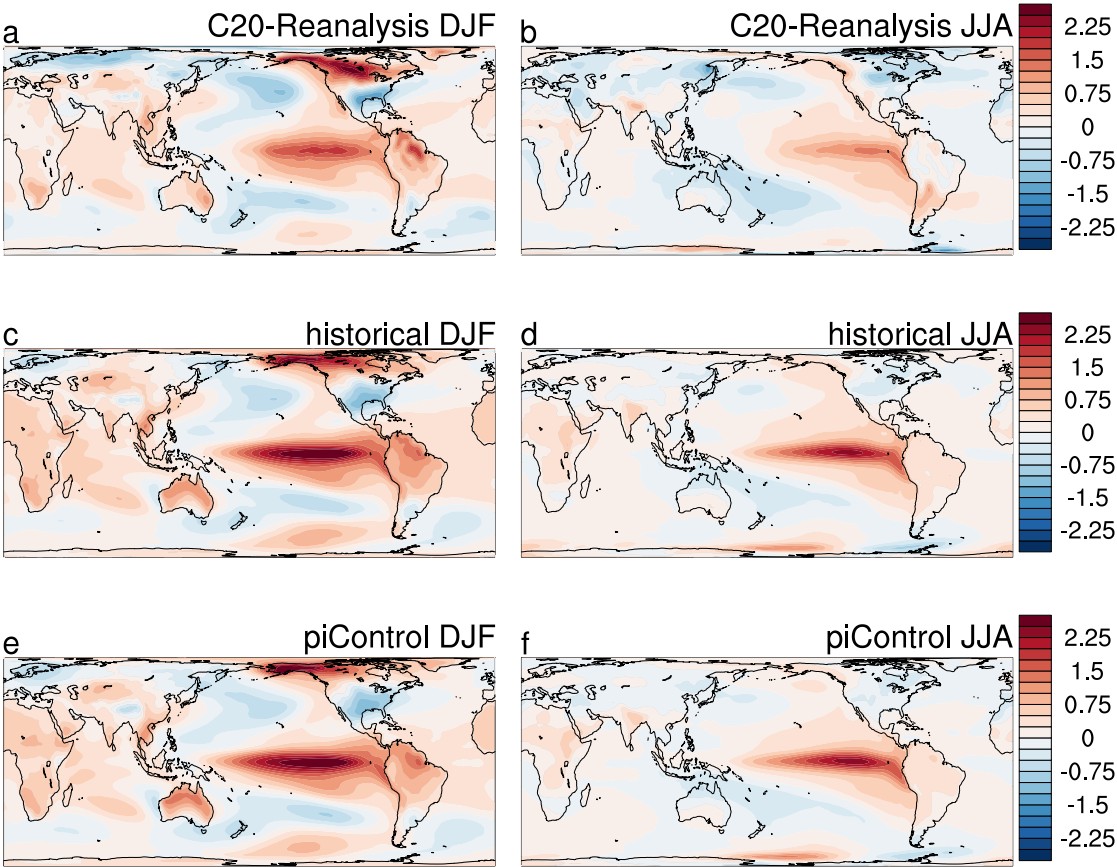

**Figure 4:** Evaluation of the ENSO temperature teleconnections. Composite El Niño minus La Niña surface temperature anomaly from (a) DJF and (b) JJA from the C20 Reanalysis (Compo et al., 2011) between 1871-2010, (c) DJF and (d) JJA model ensemble mean from the historical simulations between 1850-2005 (for CMIP5 models) or 1850-2015 (for CMIP6 models), and (e) DJF and (f) JJA model ensemble mean ensemble from the pre-industrial control simulations. Units are °C.





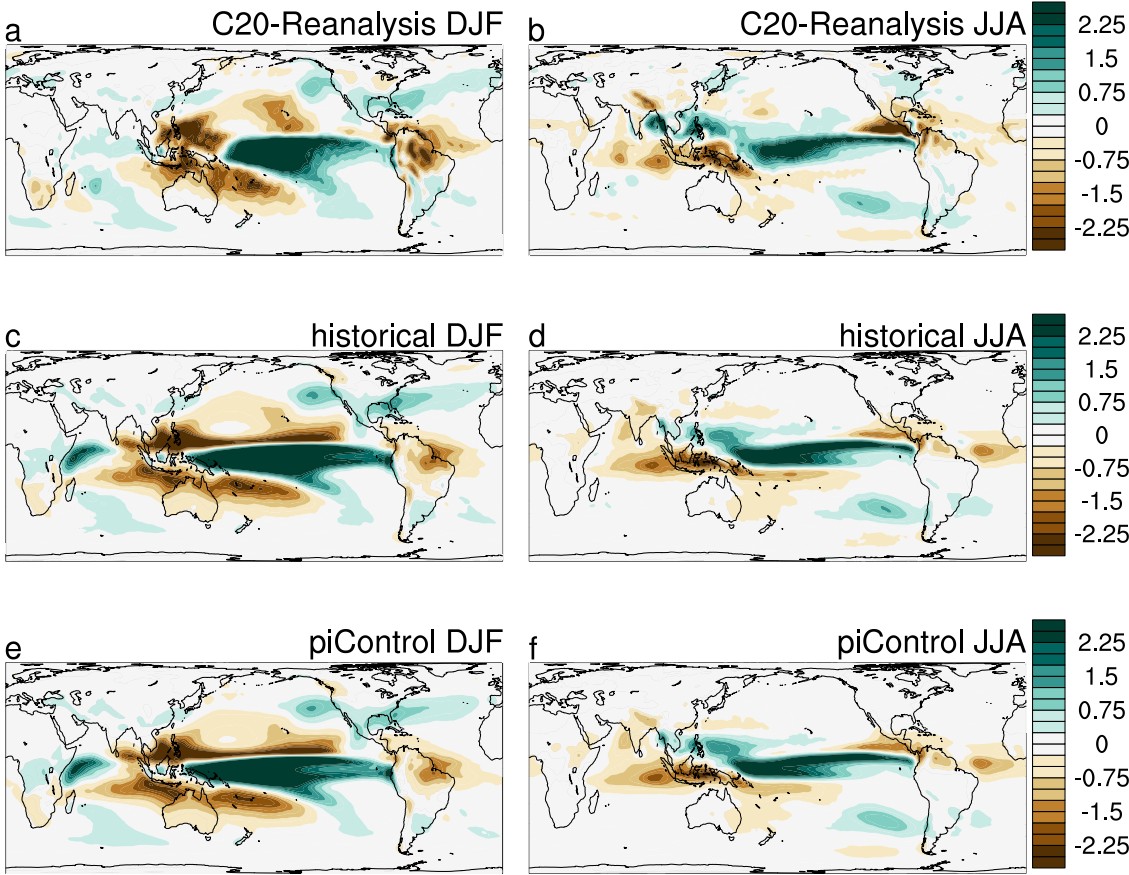

**Figure 5:** Evaluation of the ENSO precipitation teleconnections. Composite El Niño minus La Niña precipitation anomaly from (a) DJF and (b) JJA from the C20 Reanalysis (Compo et al., 2011) between 1871-2010, (c) DJF and (d) JJA model ensemble mean from the historical simulations between 1850-2005(CMIP5)/2015 (CMIP6), and (e) DJF and (f) JJA model ensemble mean ensemble from the pre-industrial control simulations. Units are mm/day.

1070





**Figure 6:** Ensemble mean seasonal changes in sea surface temperature in experiment minus pre-industrial control simulations: (a) DJF *midHolocene*, (b) JJA *midHolocene*, (c) DJF *lgm*, (d) JJA *lgm*, (e) DJF *lig127k*, (f) JJA *lig127k*, (g) DJF *1pctCO2*, (h) JJA *1pctCO2*, (i) DJF *abrupt4xCO2*, (j) JJA *abrupt4xCO2*. The ensemble mean temperature pattern in the pre-industrial control simulations is shown as black contours. Units are °C.





**Figure 7:** Ensemble mean seasonal changes in precipitation in experiment minus pre-industrial control simulations: (a) DJF *midHolocene*, (b) JJA *midHolocene*, (c) DJF *lgm*, (d) JJA *lgm*, (e) DJF *lig127k*, (f) JJA *lig127k*, (g) DJF *1pctCO2*, (h) JJA *1pctCO2*, (i) DJF *abrupt4xCO2*, (j) JJA *abrupt4xCO2*. The ensemble mean temperature pattern in the pre-industrial control simulations is shown as black contours. Units are mm/day.







**Figure 8:** Amplitude of ENSO measured from standard deviation of NINO3.4 index (°C) in *piControl* simulations (grey bars) and (a) *midHolocene* (dark green bars), (b) *lgm* (blue bars), (c) *lig127k* (light green bars), (d) *1pctCO2* (dark red bars) and (e) *abrupt4xCO2* (light red bars) simulations. Model names are given below plots.



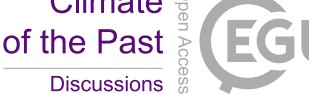

**Figure 9:** Change in amplitude of ENSO measured from standard deviation of NINO3.4 index relative to *piControl* amplitude (%) in (a) *midHolocene*, (b) *lgm*, (c) *lig127k*, (d) *1pctCO2* and (e) *abrupt4xCO2*. Model names are given below plots.



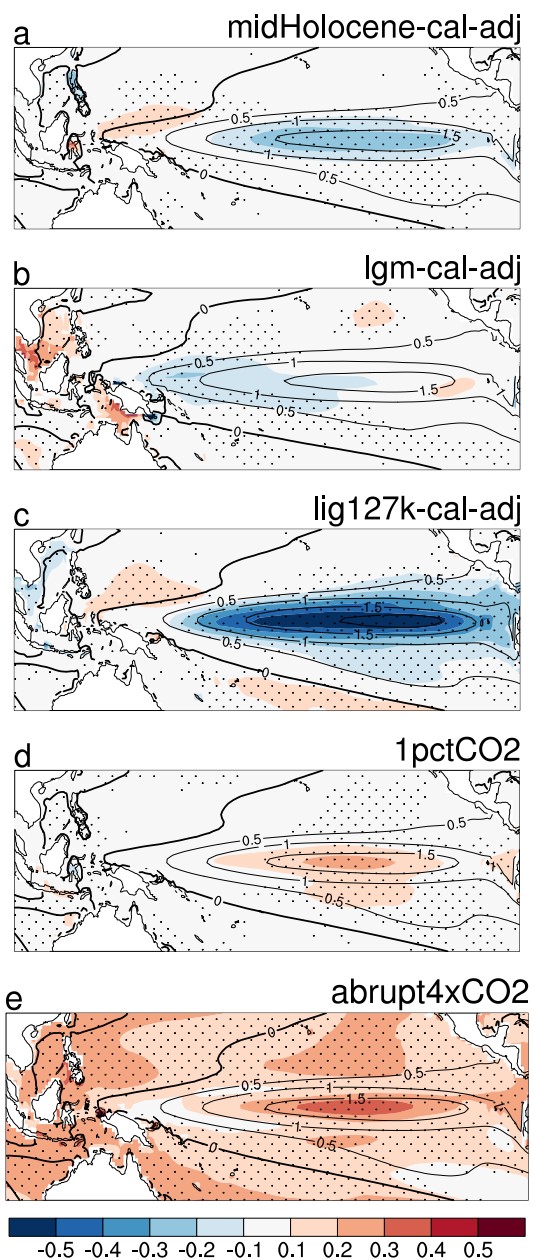

**Figure 10:** The changes in the SST pattern associated with ENSO in each experiment compared with *piControl*. The ensemble mean difference between the composites of each models El Niño minus La Niña (defined as ±1 standard deviation) in (a) *midHolocene*, (b) *lgm*, (c) *lig127k*, (d) *1pctCO2* and (e) *abrupt4xCO2* experiments minus the same pattern for the *piControl* simulations The ensemble mean ENSO SST pattern in the *piControl* simulations are shown as black contours. Stippling indicates that more than 2/3 of the ensemble members agree on the sign of the change.





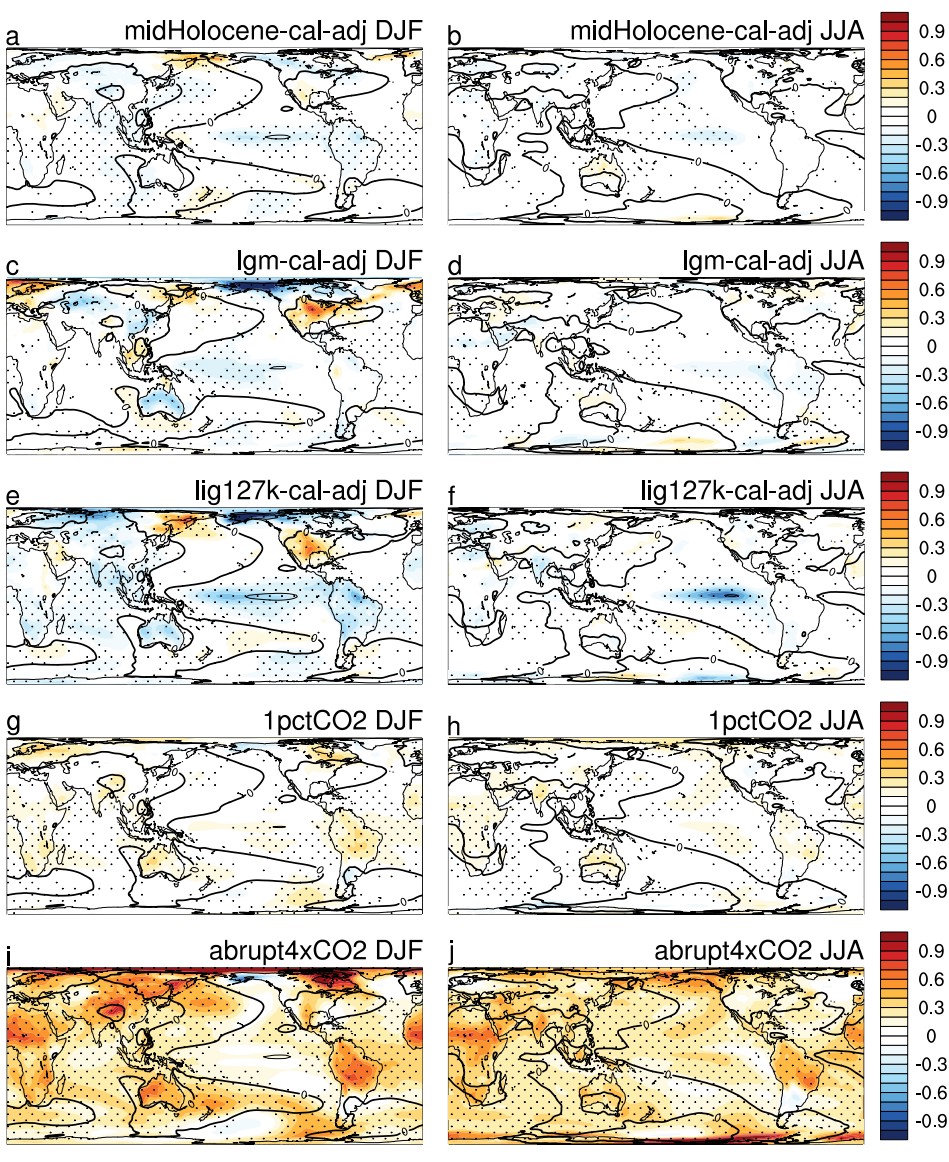

100

**Figure 11:** The changes in the seasonal temperature teleconnection pattern associated with ENSO in each experiment compared with the pre-industrial control. The ensemble mean difference between the composites of each models El Niño minus La Niña (defined as ±1 standard deviation) in (a, b) *midHolocene*, (c, d) *lgm*, (e, f) *lig127k*, (g, h) *1pctCO2* and (i, j) *abrupt4xCO2* experiments minus the same pattern for the *piControl* simulations The ensemble mean ENSO pattern in the

105  *piControl* simulations are shown as black contours. Stippling indicates that more than 2/3 of the ensemble members agree on the sign of the change.





**Figure 12:** The changes in the seasonal precipitation teleconnection pattern associated with ENSO in each experiment compared with the pre-industrial control. The ensemble mean difference between the composites of each models El Niño minus La Niña (defined as ±1 standard deviation) in (a, b) *midHolocene*, (c, d) *lgm*, (e, f) *lig127k*, (g, h) *1pctCO2* and (i, j) *abrupt4xCO2* experiments minus the same pattern for the *piControl* simulations The ensemble mean ENSO pattern in the *piControl* simulations are shown as black contours. Stippling indicates that more than 2/3 of the ensemble members agree on the sign of the change.





**Figure 13:** Relationships across all experiments and all models: (a) ENSO amplitude change (%) versus change in the annual cycle (%), (b) ENSO amplitude change (°C) versus zonal SST gradient change (°C), and (c) NINO3 precipitation for ENSO composite versus meridional SST gradient change (°C). All changes are relative to *piControl*, see text for details. Circles are coloured as follows: blue = *lgm*, dark green = *midHolocene*, light green = *lig127k*, dark red = *1pctCO2*, light red = *abrupt4xCO2* (after bar charts in Figure 8).