# Peer review of "Comparison of past and future simulations of ENSO in CMIP5/PMIP3 and CMIP6/PMIP4 models"

_Climate of the Past, 2019_

## Referee Comment (RC1) · Anonymous Referee #1 · 5 Mar 2020

Paper Quality

This paper provides a comprehensive analysis of changes in ENSO from a variety of past to future climate scenarios from CMIP5/6 and PMIP3/4 models to determine insights into the sensitivity of ENSO to changes in seasonality, global average temperatures, and spatial patterns of sea surface temperature. The authors look at past time slices from the Last Interglacial, Last Glacial Maximum, and mid-Holocene and future projections from one percent per year $CO_2$ increase and an abrupt four times $CO_2$ increase. Their choices reflect a wide range of climates in order to constrain uncertainty, determine model agreement, and evaluate sensitivity of ENSO to different climate fac-

tors. Their general findings illustrate consistent model agreement of weakened ENSO activity in the Last Interglacial and mid-Holocene, reduced variability in the western equatorial Pacific in the Last Glacial Maximum and enhanced precipitation response to ENSO in warmer $CO_2$ experiments.

Overall, the paper is well-written. The Introduction of the paper is a nice change in style from a typical climate modeling paper and was well welcomed. The authors nicely explained the historical context of ENSO, putting the Last Interglacial, Last Glacial Maximum and mid-Holocene, as well as future changes, into context of our prior knowledge from both proxies and modeling studies. Additionally, the authors produce a well-written description of the different simulations, providing clear enough detail on each of the simulations without bogging down the writing with too many technical details. The observations are clearly written out and synthesized in an easy to understand format and a thorough explanation on possible mechanisms is discussed.

Though the paper is well written, I think there are parts that need further clarification. Specifically, the authors should be clearer about the updated models compared to the previous models. At points, they suggest they will be comparing new simulations with previous generations models (I assume CMIP6/PMIP4 versus CMIP5/PMIP3?) but their results suggest that the models are more lumped together. The paper would be improved if the authors clarified when and how the newest generation models add to our understanding of changes in ENSO with respect to the simulations from previous generations models.

Scientific Questions/Issues

Lines 55-58 – There is an even newer reference from White et al., 2020 that can be added to this set of references (https://agupubs.onlinelibrary.wiley.com/doi/full/10.1029/2019GL085504)

Lines 74-84 – This paragraph could benefit from updated references. For example, Cobb et al. 2013 is now superseded by Grothe et al. (in press at Geophysical Re-

Interactive
comment

search Letters - https://agupubs.onlinelibrary.wiley.com/doi/10.1029/2019GL083906), which does show a strong reduction of ENSO variability during the mid-Holocene. Also, there's a paper by White et al. 2019 (https://agupubs.onlinelibrary.wiley.com/doi/full/10.1002/2017GL075433) that shows a long-term trend in ENSO strength through the Holocene, contradicting some of the mid-Holocene ENSO reduction studies.

Section 2.1 Models – I think it would be useful to add just a few more sentences here about the models as not all readers of this journal will have that background. This can be brief and may only include one or two sentences, and then refer the reader to the website for more information. For example, what are the main differences/improvements in CMIP6, since this paper is really about using the new generation of coupled atmosphere-ocean climate models for both past and future climates. Or maybe just more broadly, what is it about CMIP6/PMIP4 that allows for an updated view of looking at changes in ENSO? I see that the authors add little bits of this specifically sprinkled throughout section 2.2, so I think just a more broad/general description to set up the context of this study would be beneficial.

In Sections 4-6, when talking about the model ensemble and trends, it is useful to mention the model agreement. The authors do this at times, for example, on lines 338-343, but I think it would strengthen the observations if this were done more consistently throughout these sections.

Figures – note what the stippled pattern indicates in the legend (as done in Figure 11 and 12)

Technical Issues

Line 223 should read "all available data are…"

Line 407 and 410 should be Niño

Line 414 – remove parentheses around Collins et al.

Line 484 – Merryfield 2006 reference is not in the parentheses with the other references

Line 491 – Should read "This includes. . ."

---

## Referee Comment (RC2) · Anonymous Referee #2 · 5 Mar 2020

In this manuscript Brown et al. provide a summary of ENSO behavior in past and future climate simulations from the CMIP5/PMIP3 and CMIP6/PMIP4 model output available. This is a timely submission for the IPCC deadline, though I have a few concerns:

1. I can understand the framing with respect to the Pliocene but no Pliocene model output is incorporated into the discussion. This may be because the PlioMIP2 simulations were not available when this manuscript was being prepared. If this output is available now I would include it otherwise the framing of the paper is distracting. With respect to the Pliocene there are a few key citations missing. In addition to the zonal temperature gradient, several studies have characterised the thermocline conditions and

its coupling to the cold tongue (e.g. Steph et al, 2006; 2010; Ford et al. 2012; 2015). Also, a recent study by White and Ravelo (GRL, 2020) shows reduced ENSO when the thermocline is deep during the early Pliocene. They suggest mechanistically a weak thermocline feedback dampened ENSO during the Pliocene because the thermocline was deep during the Pliocene. I wouldn't use the Ford and Ravelo 2019 as evidence for ENSO. Ford and Ravelo show that the variability in the western Pacific during the Pliocene was similar to the Holocene. The ENSO variability itself is very weak in the western Pacific and the reconstructed variability largely reflects the seasonal cycle.

2. Some paleodata reconstructions that are missing from the paper introduction/discussion: LGM: Leduc et al., 2009; Koutavas and Joanides (2012); Sadekov et al., 2013; Ford et al., 2015. Mid-Holocene: White et al., 2018. Last Millennium: Rustic et al., 2015.

3. Can you expand on Line 186-187? What do you mean there has been substantial progress toward it? Given the average residence time of a deep-water parcel is 1000 years how is 150 years close to equilibrium? In what respect?

4. Lines 403-410: This paragraph is confusing to me because "mean state" can suggest different things. I usually think about it as the zonal temperature gradient rather than the eastern Pacific meridional gradient. Can you re-write this to be specific about the gradient you're referring to?

5. Line 420: The paleo reconstructions indicate the cold tongue was productive during the Pliocene (or at least similar to today, Lyle et al., 2015) and the winds were similar (Hovan 1995, Proceedings ODP)

6. Line 480: This also included a deep thermocline which is mechanistically important for ENSO. This has also been suggested for the LGM: Ford et al. 2018.

7. Figure 13 is not red green colour-blind friendly. I think for the discussion it would be useful to have two separate comparison between the CMIP5 and CMIP6 grade models.

It doesn't sound like there has been much improvement in model performance between the CMIP3 synthesis done for the IPCC report and the models here. It would be nice to know how the models are mechanistically improving or where there are known model errors.

––––––––––––––––––––––––––––––––––

---

## Referee Comment (RC3) · Anonymous Referee #3 · 26 Mar 2020

**Review of "Comparison of past and future simulations of ENSO in CMIP5/PMIP3 and CMIP6/PMIP4 models" by Brown et al.**

**Recommendation: Major revisions**

**Summary:** This study investigates the change in ENSO amplitude, ENSO related precipitation changes and ENSO teleconnections in a multi model ensemble of CMIP5/6 and PMIP3/4 models in past and future climate states. While the models agree quite well in ENSO  Holocene and LIG, the they disagree strongly in LGM, 1pctCO2 and 4xCO2.

**Overall opinion:** This is an interesting and well structured  study. The results are well elaborated and convincing. My major concern is the presentation of some SST Figures, as relative SST highlights in many cases the relation between SST change and precipitation/atmospheric circulation changes much better (see major point). A more detailed discussion of the presented results to the changes in the Walker Circulation would also allow a deeper insight into the cause of the precipitation and ENSO amplitude changes.

**Major comments:**

Introduction: You don't say anything about ENSO in the Interglacial. Please add.

Fig. 1: As for the tropical circulation the relative SST reveals the relation to precipitation and atmospheric circulation much better, please show the relative SST bias in Fig. 1c-f) and give the area mean temperature in the header  (Johnson and Xie 2010; Johnson and Kosaka 2016; Bayr et al. 2018; Izumo et al. 2019).

Fig. 6: As for Fig. 1 I would strongly suggest to show the relative SST change (and area mean SST change in the header), as this indicates the change in Walker Circulation (Bayr et al. 2014, 2020), which would be helpful to understand the precipitation change and ENSO amplitude change. Further, the change of Walker Circulation under different global mean temperatures is partly driven by the overall (homogeneous) warming (weakening under warmer and strengthening under colder mean climate, (Held and Soden 2006; Vecchi et al. 2006; DiNezio et al. 2011) and partly by the inhomogeneous warming (depends on the change of the SST gradient, Bayr and Dommenget 2013; Bayr et al. 2014). The best would be a more detailed analysis of the Walker Circulation changes to understand the ENSO amplitude change and precipitation change. But maybe you already get a clearer picture, when looking at the relative SST change.

Fig. 8 & 9: can you please show the multi model ensemble mean for each subfigure and the spread around as box plot.

Fig. 13: Please give the correlation values for each scatter plot. Further, I suggest to also look on how the wind-SST feedback changes in the scenarios, as from my experience and the study of (Vijayeta and Dommenget 2018) the change in the wind-SST feedback explain a large part of ENSO amplitude change. The change in wind-SST feedback is strongly influenced by the change in the Walker Circulation (Bayr et al. 2018, 2020).

**Minor comments:**

75: mid-Holocene – please give the years BP

120: "was replaced by the Central Pacific-type El Niño" When? At the beginning of 21$^{st}$ century? Please make clearer.

132: You should also cite here (Latif and Keenlyside 2009).

Fig. 1 & 2: What is the stippling? It is not mentioned in the figure caption.

255: "The Intertropical Convergence Zone (ITCZ) is generally shifted to the north". But also the rising branch of the Walker Circulation is shifted to the west (Bayr et al. 2018, 2020). This weakens the atmospheric feedbacks and hampers simulated ENSO dynamics. This should be discussed somewhere in the paper.

429: "must consist of several processes." An other explanation is the nonlinear behavior of ENSO amplitude and SST gradient/thermocline slope as shown in Fig. 6 in (Hu et al. 2013).

460: "increase of the negative feedback by the mean current thermal advection" Another possible factor can be an increase of the negative heat flux damping as found in (Prigent et al. 2020) for the Atlantic Nino reduction since the year 2000.

498: you should also mention here the bias ENSO dynamics due to the error compensation of the underestimated wind-SST and heat flux-SST feedback (Bayr et al. 2019) and weaker oceanic response (Kim et al. 2014).

References:

Bayr T, Dommenget D (2013) The Tropospheric Land–Sea Warming Contrast as the Driver of Tropical Sea Level Pressure Changes. J Clim 26:1387–1402. https://doi.org/10.1175/JCLI-D-11-00731.1

Bayr T, Dommenget D, Latif M (2020) Walker Circulation controls ENSO Atmospheric Feedbacks in Uncoupled and Coupled Climate Model Simulations. Clim Dyn. https://doi.org/10.1007/s00382-020-05152-2

Bayr T, Dommenget D, Martin T, Power SB (2014) The eastward shift of the Walker Circulation in response to global warming and its relationship to ENSO variability. Clim Dyn 43:2747–2763. https://doi.org/10.1007/s00382-014-2091-y

Bayr T, Latif M, Dommenget D, et al (2018) Mean-state dependence of ENSO atmospheric feedbacks in climate models. Clim Dyn 50:3171–3194. https://doi.org/10.1007/s00382-017-3799-2

Bayr T, Wengel C, Latif M, et al (2019) Error compensation of ENSO atmospheric feedbacks in climate models and its influence on simulated ENSO dynamics. Clim Dyn 53:155–172. https://doi.org/10.1007/s00382-018-4575-7

DiNezio PN, Clement A, Vecchi G a., et al (2011) The response of the Walker circulation to Last

Glacial Maximum forcing: Implications for detection in proxies. Paleoceanography 26:n/a-n/a. https://doi.org/10.1029/2010PA002083

Held IM, Soden BJ (2006) Robust responses of the hydrological cycle to global warming. J Clim 19:5686–5699. https://doi.org/10.1175/JCLI3990.1

Hu ZZ, Kumar A, Ren HL, et al (2013) Weakened interannual variability in the tropical pacific ocean since 2000. J Clim 26:2601–2613. https://doi.org/10.1175/JCLI-D-12-00265.1

Izumo T, Vialard J, Lengaigne M, Suresh I (2019) Relevance of relative sea surface temperature for tropical rainfall interannual variability. Geophys Res Lett. https://doi.org/10.1029/2019gl086182

Johnson NC, Kosaka Y (2016) The impact of eastern equatorial Pacific convection on the diversity of boreal winter El Niño teleconnection patterns. Clim Dyn 47:3737–3765. https://doi.org/10.1007/s00382-016-3039-1

Johnson NC, Xie SP (2010) Changes in the sea surface temperature threshold for tropical convection. Nat Geosci 3:842–845. https://doi.org/10.1038/ngeo1008

Kim ST, Cai W, Jin FF, Yu JY (2014) ENSO stability in coupled climate models and its association with mean state. Clim Dyn 42:3313–3321. https://doi.org/10.1007/s00382-013-1833-6

Latif M, Keenlyside NS (2009) El Niño/Southern Oscillation response to global warming. Proc Natl Acad Sci 106:20578–20583

Prigent A, Lübbecke JF, Bayr T, et al (2020) Weakened SST variability in the tropical Atlantic Ocean since 2000. Clim Dyn. https://doi.org/10.1007/s00382-020-05138-0

Vecchi GA, Soden BJ, Wittenberg AT, et al (2006) Weakening of tropical Pacific atmospheric circulation due to anthropogenic forcing. Nature 441:73–76. https://doi.org/10.1038/nature04744

Vijayeta A, Dommenget D (2018) An evaluation of ENSO dynamics in CMIP simulations in the framework of the recharge oscillator model. Clim Dyn 0:1–19. https://doi.org/10.1007/s00382-017-3981-6

---

## Author Comment (AC1) · 29 Apr 2020

General response: We appreciate the positive comments of the reviewer and the kind words written about our text and the manner in which we've approached the manuscript. We would be happy to revise the text to address the issues identified by the reviewer.

Comment: Though the paper is well written, I think there are parts that need further clarification. Specifically, the authors should be clearer about the updated models compared to the previous models. At points, they suggest they will be comparing new simulations with previous generations models (I assume CMIP6/PMIP4 versus

[Figure]

CMIP5/PMIP3?) but their results suggest that the models are more lumped together. The paper would be improved if the authors clarified when and how the newest generation models add to our understanding of changes in ENSO with respect to the simulations from previous generations models. Response: We agree that we have not discussed or quantified the differences between CMIP6 and CMIP5 model results as extensively as we should have. We will address this by including Supplementary Figures showing the two ensembles separately, as well as revising the text to highlight where the model ensembles agree on a consistent result and where the new CMIP6 models may provide new insight.

Scientific Questions/Issues Comment: Lines 55-58 – There is an even newer reference from White et al., 2020 that can be added to this set of references (https://agupubs.onlinelibrary.wiley.com/doi/full/10.1029/2019GL085504). Response: This reference will be added.

Comment: Lines 74-84 – This paragraph could benefit from updated references. For example, Cobb et al. 2013 is now superseded by Grothe et al. (in press at Geophysical Research Letters https://agupubs.onlinelibrary.wiley.com/doi/10.1029/2019GL083906), which does show a strong reduction of ENSO variability during the mid-Holocene. Also, there's a paper by White et al. 2019 (https://agupubs.onlinelibrary.wiley.com/doi/full/10.1002/2017GL075433) that shows a long-term trend in ENSO strength through the Holocene, contradicting some of the mid-Holocene ENSO reduction studies. Response: These references will be added and the discussion updated.

Comment: Section 2.1 Models – I think it would be useful to add just a few more sentences here about the models as not all readers of this journal will have that background. This can be brief and may only include one or two sentences, and then refer the reader to the website for more information. For example, what are the main differences/ improvements in CMIP6, since this paper is really about using the new gener-

ation of coupled atmosphere-ocean climate models for both past and future climates. Or maybe just more broadly, what is it about CMIP6/PMIP4 that allows for an updated view of looking at changes in ENSO? I see that the authors add little bits of this specifically sprinkled throughout section 2.2, so I think just a more broad/general description to set up the context of this study would be beneficial. Response: The discussion of models in Section 2.1 will be expanded to describe the models in more detail, and we will identify any relevant changes in CMIP6/PMIP4 generation models that may provide new information about changes in ENSO.

Comment: In Sections 4-6, when talking about the model ensemble and trends, it is useful to mention the model agreement. The authors do this at times, for example, on lines 338- 343, but I think it would strengthen the observations if this were done more consistently throughout these sections. Response: Model agreement will be included consistently in the discussion of results in Sections 4-6.

Comment: Figures – note what the stippled pattern indicates in the legend (as done in Figure 11 and 12). Response: This figure legend will be updated to add this information.

Technical Issues Comment: Line 223 should read "all available data are: : :" Response: This will be corrected. Comment: Line 407 and 410 should be Niño. Response: This will be corrected. Comment: Line 414 – remove parentheses around Collins et al. Response: This will be corrected. Comment: Line 484 – Merryfield 2006 reference is not in the parentheses with the other references. Response: This will be corrected. Comment: Line 491 – Should read "This includes: : :" Response: This will be corrected.

---

## Author Comment (AC2) · 29 Apr 2020

General response: We thank the reviewer for their thoughtful and detailed comments.

Comment 1: I can understand the framing with respect to the Pliocene but no Pliocene model output is incorporated into the discussion. This may be because the PlioMIP2 simulations were not available when this manuscript was being prepared. If this output is available now I would include it otherwise the framing of the paper is distracting. With respect to the Pliocene there are a few key citations missing. In addition to the zonal temperature gradient, several studies have characterised the thermocline conditions and its coupling to the cold tongue (e.g. Steph et al, 2006; 2010; Ford et al.

2012; 2015). Also, a recent study by White and Ravelo (GRL, 2020) shows reduced ENSO when the thermocline is deep during the early Pliocene. They suggest mechanistically a weak thermocline feedback dampened ENSO during the Pliocene because the thermocline was deep during the Pliocene. I wouldn't use the Ford and Ravelo 2019 as evidence for ENSO. Ford and Ravelo show that the variability in the western Pacific during the Pliocene was similar to the Holocene. The ENSO variability itself is very weak in the western Pacific and the reconstructed variability largely reflects the seasonal cycle. Response: The reviewer is correct in surmising that the introduction was written without a complete knowledge of the simulations that would be available for inclusion in this manuscript. As we do not include the Pliocene simulations (which are the subject of other studies), we will greatly reduce the discussion of the Pliocene in a revised manuscript. We are also very happy to update the references in the light of the reviewer's suggestions.

Comment 2: Some paleodata reconstructions that are missing from the paper introduction/ discussion: LGM: Leduc et al., 2009; Koutavas and Joanides (2012); Sadekov et al., 2013; Ford et al., 2015. Mid-Holocene: White et al., 2018. Last Millennium: Rustic et al., 2015. Response: These reference will be added, thank you for the suggestions.

Comment 3: Can you expand on Line 186-187? What do you mean there has been substantial progress toward it? Given the average residence time of a deep-water parcel is 1000 years how is 150 years close to equilibrium? In what respect? Response: We will revise the description of the abrupt4xCO2 simulations to clarify the limited progress towards equilibrium after 150 years.

Comment 4: Lines 403-410: This paragraph is confusing to me because "mean state" can suggest different things. I usually think about it as the zonal temperature gradient rather than the eastern Pacific meridional gradient. Can you re-write this to be specific about the gradient you're referring to? Response: A revised manuscript would have a rephrased paragraph about the mean state, because if it confuses this reviewer then it may also confuse the our intended readers.

Comment 5: Line 420: The paleo reconstructions indicate the cold tongue was pro-
ductive during the Pliocene (or at least similar to today, Lyle et al., 2015) and the
winds were similar (Hovan 1995, Proceedings ODP). Response: This paragraph will
be edited to reduce the focus on the Pliocene as noted in response to Comment 1.

Comment 6: Line 480: This also included a deep thermocline which is mechanistically
important for ENSO. This has also been suggested for the LGM: Ford et al. 2018.
Response: This paragraph will be edited to reduce the focus on the Pliocene as noted
in response to Comment 1. The potential role of a deep thermocline in the LGM will be
noted.

Comment 7: Figure 13 is not red green colour-blind friendly. I think for the discussion
it would be useful to have two separate comparison between the CMIP5 and CMIP6
grade models. It doesn't sound like there has been much improvement in model perfor-
mance between the CMIP3 synthesis done for the IPCC report and the models here.
It would be nice to know how the models are mechanistically improving or where there
are known model errors. Response: We apologise for not having considered the Ac-
cessibility of the figures and will be revising Figure 13 accordingly. The point raised
about the comparison between subsequent generations of CMIP models was also
raised by another reviewer and we agree that a specific paragraph on this question
should be added to the discussion.

---

## Author Comment (AC3) · 29 Apr 2020

General response: We thank the reviewer for their thorough analysis of our research.

Comment: Overall opinion: This is an interesting and well structured study. The results are well elaborated and convincing. My major concern is the presentation of some SST Figures, as relative SST highlights in many cases the relation between SST change and precipitation/atmospheric circulation changes much better (see major point). A more detailed discussion of the presented results to the changes in the Walker Circulation would also allow a deeper insight into the cause of the precipitation and ENSO amplitude changes. Response: We recognise that this work is predominantly descriptive which is unfortunately necessary given the large quantity of models and simulations being analysed here. We intend to include figures of changes in relative SST in a revised document, in addition to the existing figures of absolute SST change. We would prefer to leave evaluation of the changes in the Walker circulation for future studies as this is a substantial additional body of work that would greatly increase the size of the current manuscript.

Major comments:

Comment: Introduction: You don't say anything about ENSO in the Interglacial. Please add. Response: A revised manuscript will Include discussion of ENSO at the Last Interglacial, although there is not a large amount of research on the topic.

Comment: Fig. 1: As for the tropical circulation the relative SST reveals the relation to precipitation and atmospheric circulation much better, please show the relative SST bias in Fig. 1c-f) and give the area mean temperature in the header (Johnson and Xie 2010; Johnson and Kosaka 2016; Bayr et al. 2018; Izumo et al. 2019). Response: We will add an additional figure showing ensemble mean relative SST changes. We feel it is also important to include the absolute SST change, as they demonstrate the cooling/warming and therefore any Clausius-Clapeyron effects.

Comment: Fig. 6: As for Fig. 1 I would strongly suggest to show the relative SST change (and area mean SST change in the header), as this indicates the change in Walker Circulation (Bayr et al. 2014, 2020), which would be helpful to understand the precipitation change and ENSO amplitude change. Response: We will add an additional figure showing ensemble mean relative SST changes.

Comment: Further, the change of Walker Circulation under different global mean temperatures is partly driven by the overall (homogeneous) warming (weakening under warmer and strengthening under colder mean climate, (Held and Soden 2006; Vecchi et al. 2006; DiNezio et al. 2011) and partly by the inhomogeneous warming (depends on the change of the SST gradient, Bayr and Dommenget 2013; Bayr et al. 2014). The

best would be a more detailed analysis of the Walker Circulation changes to understand the ENSO amplitude change and precipitation change. But maybe you already get a clearer picture, when looking at the relative SST change. Response: As outlined above, we intend to include figures of changes in relative SST in a revised document. We would prefer to leave evaluation of the changes in the Walker circulation for future studies.

Comment: Fig. 8 & 9: can you please show the multi model ensemble mean for each subfigure and the spread around as box plot. Response: We will include the ensemble summary values in the bar charts, as this is a helpful addition.

Comment: Fig. 13: Please give the correlation values for each scatter plot. Response: We will include the correlation coefficient for the scatterplot figures.

Comment: Further, I suggest to also look on how the wind-SST feedback changes in the scenarios, as from my experience and the study of (Vijayeta and Dommenget 2018) the change in the wind-SST feedback explain a large part of ENSO amplitude change. The change in wind-SST feedback is strongly influenced by the change in the Walker Circulation (Bayr et al. 2018, 2020). Response: We would rather defer the analysis of the wind-SST feedback for future research which is planned on the Bjerknes Index.

Minor comments:

Comment: 75: mid-Holocene – please give the years BP. Response: This will be added.

Comment: 120: "was replaced by the Central Pacific-type El Niño" When? At the beginning of 21st century? Please make clearer. Response: This will be clarified.

Comment: 132: You should also cite here (Latif and Keenlyside 2009). Response: This will be added.

Comment: Fig. 1 & 2: What is the stippling? It is not mentioned in the figure caption. Response: The stippling will be defined in the Figure caption.

Comment: 255: "The Intertropical Convergence Zone (ITCZ) is generally shifted to the north". But also the rising branch of the Walker Circulation is shifted to the west (Bayr et al. 2018, 2020). This weakens the atmospheric feedbacks and hampers simulated ENSO dynamics. This should be discussed somewhere in the paper. Response: This will be added to the discussion.

Comment: 429: "must consist of several processes." An other explanation is the non-linear behavior of ENSO amplitude and SST gradient/thermocline slope as shown in Fig. 6 in (Hu et al. 2013). Response: This will be noted.

Comment: 460: "increase of the negative feedback by the mean current thermal advection" Another possible factor can be an increase of the negative heat flux damping as found in (Prigent et al. 2020) for the Atlantic Nino reduction since the year 2000. Response: This will be added.

Comment: 498: you should also mention here the bias ENSO dynamics due to the error compensation of the underestimated wind-SST and heat flux-SST feedback (Bayr et al. 2019) and weaker oceanic response (Kim et al. 2014). Response: This will be mentioned.

---

## Author Response (AR1)

**General comment on the revised manuscript:**

We have revised the manuscript in response to the comments of 3 reviewers. We have also taken the opportunity to update the model database to add 4 new CMIP6/PMIP4 models to the study. This required modification of Table 1 and all Figures to include the new set of models but did not substantially alter any of the conclusions of the study.

In revising the paper in response to the reviewers, we have added a new Table 2 summarising the ensemble mean ENSO variability changes for all models, CMIP5 and CMIP6 models and added Supplementary Figures S1-S4 showing results for CMIP5 and CMIP6 models separately. We have also added Supplementary Figure S5 showing temperature changes relative to the tropical mean.

**The Response to Reviewers and revised manuscript with tracked changes are provided below. 10**

**Response to Reviewer 1:**

General response: We appreciate the positive comments of the reviewer and the kind words written about our text and the manner in which we've approached the manuscript. We have revised the text in response to the comments of the reviewer, as outlined below

15

5

Comment: Though the paper is well written, I think there are parts that need further clarification. Specifically, the authors should be clearer about the updated models compared to the previous models. At points, they suggest they will be comparing new simulations with previous generations models (I assume CMIP6/PMIP4 versus CMIP5/PMIP3?) but their results suggest that the models are more lumped together. The paper would be improved if the authors clarified when and how the newest generation models add to our understanding of changes in ENSO with respect to the simulations from previous

20 generations models.

Response: We agree that we have not discussed or quantified the differences between CMIP6 and CMIP5 model results as extensively as we should have. We have addressed this by including Supplementary Figures showing the two ensembles separately, as well as revising the text to highlight where the model ensembles agree on a consistent result.

25 Scientific Questions/Issues

Comment: Lines 55-58 - There is an even newer reference from White et al., 2020 that can be added to this set of references (https://agupubs.onlinelibrary.wiley.com/doi/full/10.1029/2019GL085504)

**Response: This reference was added.**

Comment: Lines 74-84 - This paragraph could benefit from updated references. For example, Cobb et al. 2013 is now 30 superseded by Grothe et al. (in press at Geophysical Research Letters https://agupubs.onlinelibrary.wiley.com/doi/10.1029/2019GL083906), which does show a strong reduction of ENSO variability during the mid-Holocene. Also, there's a paper by White et al. 2019 (https://agupubs.onlinelibrary.wiley.com/doi/full/10.1002/2017GL075433) that shows a long-term trend in ENSO strength through the Holocene, contradicting some of the mid-Holocene ENSO reduction studies.

**Response: These references was added and the discussion updated. 35**

Comment: Section 2.1 Models - I think it would be useful to add just a few more sentences here about the models as not all readers of this journal will have that background. This can be brief and may only include one or two sentences, and then refer the reader to the website for more information. For example, what are the main differences/ improvements in CMIP6, since this paper is really about using the new generation of coupled atmosphere-ocean climate models for both past and

40 future climates. Or maybe just more broadly, what is it about CMIP6/PMIP4 that allows for an updated view of looking at changes in ENSO? I see that the authors add little bits of this specifically sprinkled throughout section 2.2, so I think just a

more broad/general description to set up the context of this study would be beneficial.

**Response:** The discussion of models in Section 2.1 was expanded to describe the models in more detail, and we identified relevant changes in CMIP6/PMIP4 generation models that may provide new information about changes in ENSO.

45 Comment: In Sections 4-6, when talking about the model ensemble and trends, it is useful to mention the model agreement. The authors do this at times, for example, on lines 338- 343, but I think it would strengthen the observations if this were done more consistently throughout these sections.

**Response:** The extent of model agreement over changes in the mean state and ENSO teleconnections are already indicated in Figures 6, 7, 10, 11 and 12 with the inclusion of stippling to show where more than 2/3 of models agree on the sign of change. No additional discussion of this aspect of model agreement was added as this would be repetitive and would make

- 50 change. No additional discussion of this aspect of model agreement was added as this would be repetitive and would make the results sections too long. The extent of model agreement over changes in ENSO amplitude (Figures 8 and 9) is already discussed in Section 5, and this was updated with the inclusion of several new models as well as Table 2 showing the multimodel mean and model range. A paragraph was added to Section 5 (and Supplementary Figure S4) to address the agreement between CMIP5 and CMIP6 models over changes in ENSO amplitude.
- 55 Comment: Figures note what the stippled pattern indicates in the legend (as done in Figure 11 and 12). Response: The figure legends were updated to add this information.

Technical Issues

Comment: Line 223 should read "all available data are: : :" Response: This was corrected.

Comment: Line 407 and 410 should be Niño. Response: This was corrected.

60 Comment: Line 414 – remove parentheses around Collins et al. Response: This was corrected.

**Comment:** Line 484 – Merryfield 2006 reference is not in the parentheses with the other references. **Response:** This was corrected (and added to reference list).

Comment: Line 491 - Should read "This includes: :: " Response: This was corrected.

**65 Response to Reviewer 2**

General response: We thank the reviewer for their thoughtful and detailed comments. We have revised the manuscript in response to the reviewer comments, as outlined below.

**Comment 1**: I can understand the framing with respect to the Pliocene but no Pliocene model output is incorporated into the discussion. This may be because the PlioMIP2 simulations were not available when this manuscript was being prepared. If this output is available now I would include it otherwise the framing of the paper is distracting. With respect to the Pliocene

this output is available now I would include it otherwise the framing of the paper is distracting. With respect to the Pilocene there are a few key citations missing. In addition to the zonal temperature gradient, several studies have characterised the thermocline conditions and its coupling to the cold tongue (e.g. Steph et al, 2006; 2010; Ford et al. 2012; 2015).

Also, a recent study by White and Ravelo (GRL, 2020) shows reduced ENSO when the thermocline is deep during the early Pliocene. They suggest mechanistically a weak thermocline feedback dampened ENSO during the Pliocene because the thermocline was deep during the Pliocene. I wouldn't use the Ford and Ravelo 2019 as evidence for ENSO. Ford and Ravelo show that the variability in the western Pacific during the Pliocene was similar to the Holocene. The ENSO variability itself

is very weak in the western Pacific and the reconstructed variability largely reflects the seasonal cycle.

Response: The Introduction has been revised to shorten the discussion of the Pliocene as we do not analyse the PlioMIP simulations (which are the subject of other studies). We have not removed all discussion of the Pliocene as we feel that it outributes to the broad context of understanding ENSO responses to past and future climate. Specifically, the first paragraph of the Introduction is a survey of the geological time-scale history of ENSO which includes the Pliocene and now also the Last Interglacial. The discussion of the Pliocene in Section 7 is now significantly shortened and only mentioned in comparison with the mid-Holocene. Reference to White and Ravelo (2020) was added.

Comment 2: Some paleodata reconstructions that are missing from the paper introduction/ discussion: LGM: Leduc et al., 2009; Koutavas and Joanides (2012); Sadekov et al., 2013; Ford et al., 2015. Mid-Holocene: White et al., 2018. Last Millennium: Rustic et al., 2015. Response: Some of these references were already included and the others have been added.

**Comment 3:** Can you expand on Line 186-187? What do you mean there has been substantial progress toward it? Given the average residence time of a deep-water parcel is 1000 years how is 150 years close to equilibrium? In what respect? **Response:** The sentence has been deleted to avoid confusion.

- 90 Comment 4: Lines 403-410: This paragraph is confusing to me because "mean state" can suggest different things. I usually think about it as the zonal temperature gradient rather than the eastern Pacific meridional gradient. Can you re-write this to be specific about the gradient you're referring to? Response: This paragraph was expanded and rewritten to more clearly explain which aspects of the mean state were examined: annual cycle, equatorial Pacific zonal SST gradient and eastern Pacific meridional SST gradient.
- 95 Comment 5: Line 420: The paleo reconstructions indicate the cold tongue was productive during the Pliocene (or at least similar to today, Lyle et al., 2015) and the winds were similar (Hovan 1995, Proceedings ODP). Response: This paragraph was edited to reduce the focus on the Pliocene as noted in response to Comment 1, therefore additional references were not added.
- Comment 6: Line 480: This also included a deep thermocline which is mechanistically important for ENSO. This has also been suggested for the LGM: Ford et al. 2018. Response: The role of the deep thermocline in suppression of ENSO variability is now mentioned in this discussion and also in the discussion of the LGM in the Introduction, with added citation of Ford et al. 2018.

Comment 7: Figure 13 is not red green colour-blind friendly. I think for the discussion it would be useful to have two separate comparison between the CMIP5 and CMIP6 grade models. It doesn't sound like there has been much improvement in model performance between the CMIP3 synthesis done for the IPCC report and the models here. It would be nice to know how the models are mechanistically improving or where there are known model errors.

**Response:** Figure 13 has been redrawn with different symbols for each experiment to ensure it is accessible. We have also added a discussion of the difference between CMIP5 and CMIP6 models throughout the paper, including 4 new supplementary figures and a new Table 2 showing changes in ENSO amplitude for CMIP5 and CMIP6 models. There is

110 actually general agreement between the two generations of models, based on the available simulations. Detailed evaluation of CMIP6 model performance in simulating ENSO is beyond the scope of this study and will no doubt be the focus of many other papers.

**Response to Reviewer 3**

115 General response: We thank the reviewer for their thorough analysis of our research.

**Comment:** Overall opinion: This is an interesting and well structured study. The results are well elaborated and convincing. My major concern is the presentation of some SST Figures, as relative SST highlights in many cases the relation between SST change and precipitation/atmospheric circulation changes much better (see major point). A more detailed discussion of the presented results to the changes in the Walker Circulation would also allow a deeper insight into the cause of the

- 120 precipitation and ENSO amplitude changes. Response: We recognise that this work is predominantly descriptive which is unfortunately necessary given the large quantity of models and simulations being analysed here. We now include figures of changes in relative SST in addition to the existing figures of absolute SST change. We would prefer to leave evaluation of the changes in the Walker circulation for future studies as this is a substantial additional body of work that would greatly increase the size of the current manuscript.
- 125 Major comments:

**Comment:** Introduction: You don't say anything about ENSO in the Interglacial. Please add. **Response:** There was already a brief discussion of proxy and model studies of ENSO in the last interglacial provided in the Discussion. A summary of these studies is now added to the Introduction as well.

Comment: Fig. 1: As for the tropical circulation the relative SST reveals the relation to precipitation and atmospheric circulation much better, please show the relative SST bias in Fig. 1c-f) and give the area mean temperature in the header (Johnson and Xie 2010; Johnson and Kosaka 2016; Bayr et al. 2018; Izumo et al. 2019). Response: The relative SST bias (with area average tropical temperature removed) is very similar to the original Figure 1, so is not included. We have produced a new plot showing the relative SST change (Fig S5) – see response to next comment.

**Comment:** Fig. 6: As for Fig. 1 I would strongly suggest to show the relative SST change (and area mean SST change in the 135 header), as this indicates the change in Walker Circulation (Bayr et al. 2014, 2020), which would be helpful to understand the precipitation change and ENSO amplitude change. **Response:** We have produced a new plot showing the relative SST change (a new version of Figure 6) which is included as Supplementary Figure S5. The absolute SST change is retained as Figure 6.

- Comment: Further, the change of Walker Circulation under different global mean temperatures is partly driven by the overall (homogeneous) warming (weakening under warmer and strengthening under colder mean climate, (Held and Soden 2006; Vecchi et al. 2006; DiNezio et al. 2011) and partly by the inhomogeneous warming (depends on the change of the SST gradient, Bayr and Dommenget 2013; Bayr et al. 2014). The best would be a more detailed analysis of the Walker Circulation changes to understand the ENSO amplitude change and precipitation change. But maybe you already get a clearer picture, when looking at the relative SST change. Response: As outlined above, we now include the relative SST
- 145 change as a new Supplementary Figure S5, and discuss this in the Section 4. We would prefer to leave evaluation of the changes in the Walker circulation for future studies.

**Comment:** Fig. 8 & 9: can you please show the multi model ensemble mean for each subfigure and the spread around as box plot. **Response:** We now include the multi-model mean values in Figure 8 and 9 and also in the new Table 2. Table 2 also includes the maximum and minimum model ENSO change values for each experiment.

150 Comment: Fig. 13: Please give the correlation values for each scatter plot. Response: We now include the correlation coefficients for the scatterplot figures (and the text in the final paragraph of Section 6 was modified slightly to refer to these correlations).

Comment: Further, I suggest to also look on how the wind-SST feedback changes in the scenarios, as from my experience and the study of (Vijayeta and Dommenget 2018) the change in the wind-SST feedback explain a large part of ENSO

155 amplitude change. The change in wind-SST feedback is strongly influenced by the change in the Walker Circulation (Bayr et al. 2018, 2020). Response: We would rather defer the analysis of the wind-SST feedback for future research which is planned on the Bjerknes Index.

Minor comments:

Comment: 75: mid-Holocene - please give the years BP. Response: Done.

160 Comment: 120: "was replaced by the Central Pacific-type El Niño" When? At the beginning of 21st century? Please make clearer. Response: This is clarified: in recent decades.

Comment: 132: You should also cite here (Latif and Keenlyside 2009). Response: This was added.

**Comment:** Fig. 1 & 2: What is the stippling? It is not mentioned in the figure caption. **Response:** The stippling is now defined in the Figure caption.

165 Comment: 255: "The Intertropical Convergence Zone (ITCZ) is generally shifted to the north". But also the rising branch of the Walker Circulation is shifted to the west (Bayr et al. 2018, 2020). This weakens the atmospheric feedbacks and hampers simulated ENSO dynamics. This should be discussed somewhere in the paper. Response: As this section is focused on precipitation biases in the models, we do not directly examine the atmospheric circulation. However we added a brief mention of the displacement of the Walker circulation due to SST biases here.

170 Comment: 429: "must consist of several processes." An other explanation is the nonlinear behavior of ENSO amplitude and SST gradient/thermocline slope as shown in Fig. 6 in (Hu et al. 2013). Response: This is now noted.

**Comment:** 460: "increase of the negative feedback by the mean current thermal advection" Another possible factor can be an increase of the negative heat flux damping as found in (Prigent et al. 2020) for the Atlantic Nino reduction since the year 2000. **Response:** This is an interesting suggestion but investigation of this mechanism is beyond the scope of the current

175 study.

**Comment:** 498: you should also mention here the bias ENSO dynamics due to the error compensation of the underestimated wind-SST and heat flux-SST feedback (Bayr et al. 2019) and weaker oceanic response (Kim et al. 2014). Response: These reference were added.

**Comparison of past and future simulations of ENSO in **CMIP5/PMIP3 and CMIP6/PMIP4 models**

Josephine R. Brown1, Chris M. Brierley2, Soon-Il An3, Maria-Vittoria Guarino4, Samantha Stevenson5, Charles J. R. Williams6,7, Qiong Zhang8, Anni Zhao2, Pascale Braconnot9, Esther C. Brady10, Deepak Chandan11, Roberta D'Agostino12, Chuncheng Guo13, Allegra N. LeGrande14, Gerrit Lohmann15, Polina 185 A. Morozova16, Rumi Ohgaito17, Ryouta O'ishi18, Bette L. Otto-Bliesner10, W. Richard Peltier11, Xiaoxu Shi15, Louise Sime4, Evgeny M. Volodin19, Zhongshi Zhang13, and Weipeng Zheng20

1School of Earth Sciences, University of Melbourne, Parkville, VIC, 3010, Australia

190 2Department of Geography, University College London, London, WCIE 6BT, UK 3Department of Atmospheric Sciences, Yonsei University, Seoul, Korea 4British 
[revised manuscript text omitted]

| (             | Deleted: underlying                              |
|---------------|--------------------------------------------------|
| ~(            | Deleted: s                                       |
| $\mathcal{A}$ | Deleted: aspects of the mean state and           |
|               |                                                  |
| (             | Deleted: no significant                          |
| ~(            | Deleted: also                                    |
| (             | Deleted: extreme El Nino and the mean state      |
| (             | Deleted: mean state is represented as changes in |
| ~(            | Deleted: -                                       |

W–90° W). The strength of eastern Pacific El Niño rainfall is represented as the changes in ENSO composite precipitation over the NINO3 region (5° S–5° N, 150° W–90° W) normalised by the NINO3.4 standard deviation used to identify the composited events. This normalisation aims to remove the impact of the changes in ENSO variability documented between the experiments (e.g. Power and Delage, 2018). The significant negative correlation is consistent with the analysis demonstrated by Cai et al. (2014) and Collins et al. (2019), but this analysis approach allows the relationship to be visualised across many more simulations and experiments. This relationship appears to be fundamental feature of ENSO behaviour, rather than just a response to greenhouse gas forcing.

**7 Mechanisms and Discussion**

- There is evidence that the mid-Holocene, a period of supressed ENSO variability, featured a stronger zonal gradient in the tropical Pacific mean SST than the 20th century, namely "La Niña-like conditions" (Barr et al., 2019; Gagan and Thompson, 2004; Koutavas et al., 2002; Luan et al., 2012; Shin et al., 2006). In contrast, proxy records suggest that the mean state of the tropical Pacific during the Pliocene warm period featured sustained El Niño-like conditions (Fedorov et al., 2006; Wara et al., 2005). It is likely that the weak zonal SST gradient in the Pliocene was less favourable for ENSO occurrence (Brierley, 2013; Manucharyan and Fedorov, 2014). These contradictory responses imply that the dynamical mechanisms determining
- 670 the relationship between the zonal gradient in mean SST and ENSO amplitude (e.g., Sadekov et al., 2013) must consist of several processes. The relationship between ENSO amplitude and SST gradient may also be nonlinear (Hu et al., 2013). This lack of consistent or linear relationship between zonal SST gradient and ENSO amplitude is supported by the results presented here, shown in Figure 13b.
- During the mid-Holocene, the reduced tropical insolation led to the cooling of the tropical Pacific, directly producing a La Niña-like condition. Under La Niña-like conditions, the air-sea coupling strength is reduced due to a suppressed convective instability, and thus ENSO variability is suppressed (Liu et al., 2000; Roberts et al., 2014). The stronger seasonality in insolation over the Northern Hemisphere associated with the precession cycle resulted in a stronger annual cycle, which could also act to reduce ENSO variability through the intensified annual-frequency entrainment (Liu, 2002; Pan et al., 2005). A similar but stronger precession effect due to the higher eccentricity during the last interglacial period was also found to
- 680 have a relatively weak ENSO amplitude in palaeo-proxy records (Hughen et al., 1999; Tudhope et al., 2001) and climate model simulations (An et al., 2017; Salau et al., 2012). However, the mid-Holocene simulations of PMIP2/3 mostly showed a reduction of both annual cycle and ENSO amplitude (An and Choi, 2014; Masson-Delmotte et al., 2014).

The reduced annual cycle over the tropical eastern Pacific is attributed to the relaxation of eastern Pacific upper-ocean stratification due to the annual downwelling Kelvin wave forced by western Pacific wind anomalies (Karamperidou et al.,

685 2015) or the deepening of ocean mixed layer depth associated with the northward shift of the ITCZ (An and Choi, 2014). Therefore, the mid-Holocene ENSO variability in PMIP2/3 may be deemed to the result of the counterbalance between the

**20**

**Deleted: n**

| (                 | Deleted: strong negatively |
|-------------------|----------------------------|
| (                 | Deleted: relationship      |
| ~~(               | Deleted: (                 |
| $\mathbb{X}$      | Deleted: ,                 |
| $\langle \rangle$ | Deleted:                   |
| Ì                 | Deleted: to that           |
|                   |                            |

Deleted: The mean state of the tropical Pacific during the Pliocene warm period featured sustained El Niño-like conditions (Fedorov et al., 2006; Wara et al., 2005). Such weak zonal gradient in mean equatorial Pacific SST and the deep eastern Pacific thermoeline are linked to strongly suppressed trade winds. The suppressed trade winds, or westerly so-called 'super-rotation flow', can be driven by the equatorward westerly eddy momentum flux originating from the enhanced poleward Rossby wave pumping due to the stronger tropical convection under the warm ocean surface (Arnold et al., 2012; Tziperman and Farrell, 2009). Whilst individual El Niño events occurred during the Pliocene warm period (Ford and Ravelo, 2019), it is likely that the weak zonal contrast in mean SST (i.e., El Niño-like mean condition) was less favourable for ENSO occurrence (Brierley, 2013; Manucharyan and Fedorov, 2014). ...

| Deleted: | another                  |
|----------|--------------------------|
| Deleted: | suppression period       |
| Deleted: | annual cycle             |
| Deleted: | between the Pliocene and |
| Deleted: | mid-Holocene             |
| Deleted: |                          |

[revised manuscript text omitted]
 10.3) | =                             | Ξ.                            | -20.6                         | 12                            |
| lpctCO2 | 5.6 (-33.2 to 34.1)   | 1.4                    | 13                     | 11.6                   | 9                             |
| abrupt4xCO2    | 2.2 (-46.3 to 57.7)   | 3.0                    | 12                     | 1.4                    | 11                            |

---

## Author Response (AR2)

**Response to Editor's Comments:**

*Editor's Comment 1: Firstly, many of the figures use stippling as a means to indicate model consensus. While this convention is widely applied in the climate community, I would argue that it is extremely misguided: it obscures areas of consensus (which are presumably of greater interest) and focuses the eye on the rest of the field (which is presumably of lesser interest, i.e. carrying a less robust signal). I would therefore encourage you to flip this convention, stippling the areas of model disagreement. If you need a precedent, you may find one in Hu, J., J. Emile-Geay, and J. Partin (2017), Correlation-based interpretations of paleoclimate data – where statistics meet past climates, Earth and Planetary Science Letters, 459, 362–371, doi:10.1016/j.epsl.2016.11.048. Just because a convention is widely applied, doesn't make it legitimate, and I have yet to hear anyone articulate a clear rationale for it.*

**Response:** While we agree with the Editor that there may be a case to apply stippling to areas of model disagreement, we do not think that it would be beneficial to modify the figures in this paper to reverse the stippling. The stippling of areas of model agreement is widely used, including in IPCC reports and in the wider CMIP community. This manuscript is intended to summarise new model results in an accessible and straightforward manner, and allow comparison with previous studies, including the IPCC reports. We are concerned that using a relatively novel and uncommon form of stippling may confuse readers. We also feel that the figures are not obscured in the regions of stippling as a small "dot size" is used. **Changes:** None.

*Editor's Comment 2: Secondly, the definition of ENSO amplitude is rather vague, and as such open to misinterpretation. As far as I can tell, it is the raw standard deviation of the NINO3.4 index, without any kind of bandpass-filtering applied. This is dangerous, as it lumps all timescales together, including trends (if they are present). It would be more rigorous to apply a 2-8 year bandpass filter, to make sure that the statistic in question is not capturing decadal, multidecadal, or longer, variability. This seems like an appropriate safeguard given that the manuscript focuses on ENSO, not lower-frequency tropical Pacific variability.*

**Response:** The paper defines the ENSO amplitude as the standard deviation of the NINO3.4 index (in Section 5), which is a common definition used in a range of studies including the previous IPCC report (e.g. AR5 WG1 Figures 5.10 and 14.14). The NINO3.4 timeseries are detrended to ensure the removal of long-term linear trends. We find that filtering using a 2-8 year band-pass filter (see Review Figure A below) produces ENSO amplitudes that are much lower than unfiltered values (Figure 8) and are therefore difficult to compare with previous studies such as IPCC reports. Further, the percentage changes in ENSO amplitudes using band-pass filtered NINO3.4 indices (see Review Figure B below) are very similar to the unfiltered changes (Figure 9). **Changes:** We have added a note discussing our choice not to use band-pass filtering and noting the similar fractional change for filtered and unfiltered time series at the end of Section 5, and now include the filtered ENSO amplitude change figure as Supplementary Figure S5 for comparison.

[Figure]

**Review Figure A:** ENSO amplitude as in Figure 8 but using 2-8 year band-pass filtered NINO3.4 indices.

[Figure]

**Review Figure B:** ENSO amplitude change as in Figure 9 but using 2-8 year band-pass filtered NINO3.4 indices. [Note: this Figure is also included as Supplementary Figure S5].

40   **The manuscript as revised due to Editor's comments with tracked changes is included below. Note that there have also been several minor corrections to affiliations and acknowledgements in this version of the manuscript.**

[revised manuscript text omitted]